

# Multi-objective genetic programming strategies for topic-based search with a focus on diversity and global recall

Cecilia Baggio[1,2], Carlos M. Lorenzetti[1,2], Rocío L. Cecchini[1,2] and Ana G. Maguitman[1,2]

[1] Instituto de Ciencias e Ingeniería de la Computación (UNS-CONICET), Bahia Blanca, Buenos Aires, Argentina
[2] Departamento de Ciencias e Ingeniería de la Computación, Universidad Nacional del Sur, Bahia Blanca, Buenos Aires, Argentina

## ABSTRACT

Topic-based search systems retrieve items by contextualizing the information seeking process on a topic of interest to the user. A key issue in topic-based search of text resources is how to automatically generate multiple queries that reflect the topic of interest in such a way that precision, recall, and diversity are achieved. The problem of generating topic-based queries can be effectively addressed by Multi-Objective Evolutionary Algorithms, which have shown promising results. However, two common problems with such an approach are loss of diversity and low global recall when combining results from multiple queries. This work proposes a family of Multi-Objective Genetic Programming strategies based on objective functions that attempt to maximize precision and recall while minimizing the similarity among the retrieved results. To this end, we define three novel objective functions based on result set similarity and on the information theoretic notion of entropy. Extensive experiments allow us to conclude that while the proposed strategies significantly improve precision after a few generations, only some of them are able to maintain or improve global recall. A comparative analysis against previous strategies based on Multi-Objective Evolutionary Algorithms, indicates that the proposed approach is superior in terms of precision and global recall. Furthermore, when compared to query-term-selection methods based on existing state-of-the-art term-weighting schemes, the presented Multi-Objective Genetic Programming strategies demonstrate significantly higher levels of precision, recall, and F1-score, while maintaining competitive global recall. Finally, we identify the strengths and limitations of the strategies and conclude that the choice of objectives to be maximized or minimized should be guided by the application at hand.

# INTRODUCTION

A topic-based search is the task of exploring and suggesting relevant content related to a specific subject of interest. In a topic-based search system, a topic description represents

Corresponding authors
Cecilia Baggio, cb@cs.uns.edu.ar
Ana G. Maguitman,
agm@cs.uns.edu.ar

the user's interests. A collection of labeled documents is used to characterize the relevant content for that topic and hence used for training the system. The process of suggesting material based on a topic differs from classical search in that instead of fulfilling a specific consultation need (*e.g.*, using a single query to answer a specific question such as "what is the time in London?") it aims at collecting topic-relevant resources (*e.g.*, using multiple queries to retrieve as many documents as possible relevant to a specific topic such as "solar system"). Topic-based search can be used to support total-recall retrieval (*Abualsaud et al., 2018*), knowledge extension (*Lorenzetti et al., 2016*), search for topic-relevant resources (*Hyung, Park & Lee, 2017*; *Hiriyannaiah et al., 2020*; *Scells et al., 2020*), build vertical portals (*Pinho, Franco & Mendes, 2018*), and generate topic-based alerts (*Eckroth & Schoen, 2019*), among other applications.

Query generation has an important effect on information retrieval-related tasks, particularly in the topic-based search scenario (*Culpepper et al., 2021*). A central challenge for topic-based search is to formulate multiple effective queries to achieve high coverage of relevant results. This will ensure high global recall, *i.e.*, high recall when combining the results retrieved by the individual queries. Assume the user has already gathered a small set of relevant content (*e.g.*, scientific articles, tweets, news articles, *etc.*) related to the topic of interest and would like to retrieve all of the relevant material from a particular source (*e.g.*, all the relevant scientific articles from a digital repository, all the relevant tweets, or all the relevant news article). This problem is particularly hard if the user knows little about the topic being researched as it will be very difficult to come up with meaningful vocabulary for query generation (*Culpepper et al., 2021*). In light of these difficulties, the user can be assisted by a tool that automatically generates queries based on the topic at hand. Automatic query generation involves the selection of relevant terms to construct topic-based queries and ensuring the correct query structure. Several methods for topic-based query generation were proposed in previous research works (*Budzik, Hammond & Birnbaum, 2001*; *Cecchini et al., 2008*; *Lorenzetti & Maguitman, 2009*; *Cecchini et al., 2010, 2018*; *Maisonnave et al., 2021*).

Topic-based search is different from the most widely known *ad-hoc* retrieval task (*Ma et al., 2021*). Note that the purpose of *ad-hoc* retrieval is to obtain relevant documents in response to a query specified by a user. In general, *ad-hoc* retrieval requires high precision while recall and diversity are less important. On the other hand, topic-based search focuses on retrieving relevant documents in response to an information need specified by a topic description. Different from *ad-hoc* retrieval, the goal of topic-based search is to obtain as many relevant documents as possible for further analysis. While precision is important in topic-based search, the emphasis is placed on attaining high global recall and diversity. Topic-based search usually relies on automatic query formulation. Hence, one of the most challenging problems in topic-based search is to automatically formulate queries to retrieve material that is simultaneously relevant and diverse.

The goal of this article is to propose computational strategies aimed at automating the process of topic-based search. In the proposed framework the task of topic-based search is formulated as a supervised machine learning problem. Learning to generate queries based on a given topic can be seen as an optimization problem. The objective is to maximize the

effectiveness of these queries using a training set containing documents labeled as either relevant or irrelevant. This optimization problem is characterized by several noteworthy aspects. Firstly, the solutions consist of queries that can be created using various combinations of terms, resulting in a search space of high dimensionality. Secondly, the solutions cannot be efficiently constructed from optimal solutions of its subproblems, as an effective query may consist of terms that are ineffective as single-term queries. Thirdly, multiple quasi-optimal solutions may be desirable instead of a single optimal one. Lastly, the problem involves several potentially conflicting objectives, resulting in a Multi-Objective Optimization Problem (MOOP) that requires balancing multiple criteria, such as precision, recall, or other metrics. Given these characteristics, evolutionary algorithms (EAs), specifically Multi-Objective Evolutionary Algorithms (MOEAs), provide a natural approach to addressing this optimization problem (*Boussaïd, Lepagnot & Siarry, 2013*; *Deb, 2015*; *Huang, Zhang & Li, 2019*; *Coello et al., 2019*). Previous research (*Cecchini et al., 2008*, *2010*) has successfully employed MOEAs to tackle this optimization problem, utilizing precision and recall of the individual queries as objectives to be maximized. However, for many topical-search applications, queries should be evaluated collectively rather than independently from each other. This means that we must not only strive for high performance for each specific query, but it is also imperative to ensure sufficient diversity and recall across the entire population. These challenges present a novel optimization scenario wherein preserving diversity and achieving high global recall are key objectives.

This work introduces a family of novel Multi-Objective Genetic Programming (MOGP) strategies that attempt to simultaneously attain topical relevance and diversity. MOGP is a form of MOEA where individuals representing potential solutions are encoded by means of rich structures (typically tree structures). For instance, an individual can encode a computer program. In this work, each individual encodes a tree-structured Boolean query. The strategies analyzed in this work were implemented using the DEAP platform (*Fortin et al., 2012*). The data and full code of the methods used by the framework and to carry out the experiments are made available to allow reproducibility (https://github.com/ceciliabaggio/mogp_with_terms, https://github.com/ceciliabaggio/term_weighting_methods_full_testing_dataset, https://data.mendeley.com/datasets/9mpgz8z257/1).

The main innovative features of the proposed strategies are the use of genetic programming (GP), representing each individual as a tree-structured Boolean query, and the introduction of novel fitness functions. These fitness functions consider query performance, taking into account not only the results obtained by individual queries but also the results returned by other queries within the same population. Two of the new fitness functions are reformulations of the classical precision and recall metrics and we refer to them as entropic precision and entropic recall, respectively. Another performance metric uses the Jaccard similarity index to measure the similarity between the set of results retrieved by a query and those returned by the other queries in the population. The objective of using these three metrics is to promote diversity by imposing penalties on queries that retrieve outcomes already obtained by other queries in the same population. The proposed metrics are combined in different ways among themselves and with other

traditional metrics to favor precision, global recall, and diversity simultaneously, resulting in different computational strategies for topical search. The evaluation of the strategies implemented involves measuring the classical precision and recall metrics, as well as utilizing custom metrics designed to evaluate the diversity and global recall of relevant results retrieved by the entire population of queries.

The main contribution of this work is the proposal and evaluation of new MOGP strategies for topic-based search. An important difference between the work proposed here and existing related work is the focus on dealing with diversity from an integral point of view, _i.e._, our work not only focuses on evaluating the objective functions of each individual query but also on evaluating the diversity and novelty of the recovered material when compared with the material recovered by the rest of the queries.

### Research questions

This work describes and evaluates a framework that addresses the problem of formulating topical queries when multiple objectives need to be balanced. The analysis of the proposed framework is guided by the following research questions:

**RQ1** How effective are the queries evolved by each of the proposed strategies in comparison to the initial queries that are formulated directly from the description of the topic?

**RQ2** Are the queries evolved by the proposed strategies effective when tested for the same topic but on a new _corpus_? In other words, are the proposed strategies able to avoid overfitting?

**RQ3** What are the trade-offs among different MOGP strategies for topical search when attempting to address conflicting objectives such as high precision and recall?

**RQ4** How effective are the queries evolved by the proposed strategies in comparison to queries that are built using MOEA strategies?

**RQ5** How effective are the queries evolved by the proposed strategies in comparison to queries that are built using state-of-the-art term-weighting schemes?

**RQ6** If there are significant improvements in terms of effectiveness, then we aim at finding out if the computational time required is justifiable based on the effectiveness achieved.

## BACKGROUND

### Topic-based search

Topic-based search is the process of seeking and suggesting material based on a topic of interest, which is usually defined as a topical context (_Leake & Scherle, 2001_; _Maguitman, 2018_). Topical contexts may be obtained from the user's long-term or current interests and can be defined at different levels of granularity. For instance, a topical context may be defined as a collection of documents related to a particular subject of interest. Alternatively, a user may highlight a few words in a document with the purpose of requesting information related to the highlighted text. This gives rise to a more focused view of the topical context, which may include the sentence containing the highlighted

text, the paragraph where the highlighted text resides, or the whole document the user is reading or editing. The topic of interest is initially captured as raw data and may include terms, stems, n-grams, or phrases.

The process of searching for online data can be guided by diverse objectives. There are essential differences between searching for information to fulfill a consultation need and seeking material to support the process of topic-based search of related material. Usually, the purpose of a consultation is to find specific answers to specific questions. In this case, it is crucial to attain high precision. On the other hand, when collecting resources to support topic-based search, rather than a specific question there is typically a topic of interest that can be used to initialize and refine the relevant search space (*Barathi & Shanmugam, 2014*). In this case, while precision is important, other criteria should be considered, such as high recall, novelty, and diversity.

With the advent of robust search interfaces, it has become feasible to create a plethora of applications focused on topic-based exploration. In many circumstances, as it is the case of the Deep Web and other dynamically generated content, the only access point to relevant material is through the submission of queries. As a result, acquiring the ability to automatically generate efficient topical queries has emerged as a crucial research challenge. The effectiveness of a query depends on what the user is trying to achieve. If the goal is to attain good coverage then diverse queries will be more effective than focused ones. In this case, the user will try to retrieve most of the relevant material, albeit diluted with irrelevant answers. On the other hand, if the goal is to attain good precision, specific queries will be preferred over broad and diverse ones.

Since a topic representation may contain numerous terms, selecting useful terms is crucial. Several studies have shown the benefits of having tools that provide assistance for topic-based query formulation (and reformulation) (*Culpepper et al., 2021*). Typically, these systems provide a browsing interface where the user intervention must be explicit. However, the burden implied by the need to explicitly formulate search requests can be alleviated if queries are produced automatically based on the topic representation (*Budzik, Hammond & Birnbaum, 2001*; *Chang, Ounis & Kim, 2006*; *Cecchini et al., 2008*; *Lorenzetti & Maguitman, 2009*; *Cecchini et al., 2010*, *2018*). While not all the information contained in a topic representation can be summarized in a query, effective mechanisms can be designed to extract small sets of representative terms to construct queries. While the automatic generation of queries offers a convenient mechanism to reflect a topic in the search process, the use of a topic representation as a source of queries only offers a partial solution to the problem of topic-based search. If queries are built as a conjunction of terms, short queries will only partially represent the topic at hand, resulting in poor precision, while long queries will become too specific, resulting in poor recall. On the other hand, disjunctive queries will typically result in poor precision for both short and long queries. Learning to formulate queries with more complex syntaxes, such as general Boolean queries, may result in better specifications of the user requirements (*Cordón, Herrera-Viedma & Luque, 2002*).

While Boolean operators enable the creation of highly expressive queries, many search interfaces impose limitations on query length. Consequently, formulating shorter queries

offers a more widely applicable solution compared to constructing lengthy ones. Furthermore, our objective involves identifying diverse suboptimal solutions that capture different aspects of the topic being analyzed, as opposed to relying on a single, optimal query that represents the target topic or several subtopics simultaneoulsy. Moreover, the availability of diverse queries within the population, rather than just a single query, makes the application of an evolutionary approach possible.

## Multi-objective evolutionary algorithms

EAs (*Holland, 1975*; *Eiben & Smith, 2015*) are robust optimization techniques based on the principle of "natural selection and survival of the fittest", postulated by Charles Darwin, a precursor of the scientific literature and the foundation of the theory of evolutionary biology. According to this principle "in each generation, the stronger individual survives and the weaker dies." Therefore, each new generation would contain stronger (fitter) individuals than their ancestors.

To use EAs in optimization problems we need to define candidate solutions (also called individuals) by chromosomes consisting of genes and a fitness function to be minimized (or maximized). A population of individuals is maintained and the goal is to obtain better solutions after some generations. To produce a new generation, EAs typically use selection together with the genetic operators of crossover and mutation. Parents are selected to produce offspring, favoring those parents with the highest values of the fitness function. The crossover of population members takes place by exchanging subparts of the parent chromosomes (roughly mimicking a mating process), while mutation is the result of a random perturbation of the chromosome (*e.g.*, replacing the value of a gene with another allele).

Although crossover and mutation can be implemented in many different ways, their fundamental purpose is to explore the search space of candidate solutions (also known as decision variable space), improving the population at each generation by adding better offspring and removing inferior ones. It is important to remark that this exploration is guided by the selection operator, which uses the fitness values associated with the candidate solutions. Therefore, these fitness values constitute the space where the selection works (also known as objective function space). For this reason, the fitness function defines a mapping between the decision variable space and the objective function space.

In MOOPs the quality of a solution is defined by its performance in relation to several, possibly conflicting, objectives. Traditional methods are very limited because, in general, they become too expensive as the size of the problem grows (*Lin & He, 2005*). In this context, MOEAs became suitable techniques for dealing with MOOPs (*Deb, 2014*; *Eiben & Smith, 2015*; *Coello Coello, Lamont & Van Veldhuizen, 2007*). There are many approaches to multi-objective optimization using MOEAs and, in general, they can be classified as Pareto or non-Pareto EAs. In the first case, the evaluation is made following the Pareto dominance concept (*Pareto, 1896*).

Dominance is a partial order that could be established among vectors defined over an $n$-dimensional space. By means of a multi-objective fitness function, it is possible to define a relation between vectors $x_i$ in the decision variable space and vectors $u_i$ in the objective

function space. Therefore, a Pareto optimal solution is an individual, $x_j$, within the decision variable space, whose corresponding vector components in the objective function space, $u_j$, cannot be all simultaneously improved for any other $u_i$, with $i \neq j$. A non-dominated set of a feasible region in the objective function space defines a Pareto Front and the set of its associated vectors in the decision variable space is called the Pareto Optimal Set. The Pareto-based algorithms use the concept of domination for the selection mechanism to move a population toward the Pareto Front. The results discussed in this work are based on the application of NSGA-II (Non-dominated Sorting Genetic Algorithm-II) proposed by *Deb et al. (2002)*, which is one of the most studied and efficient Pareto EAs and is based on non-domination sorting.

NSGA-II is one of the most studied and efficient MOEAs, consequently it was used in this work. The algorithm begins by creating a random parent population $P_0$ of size $n$. The population is sorted based on the non-domination concept. Each solution is assigned a rank equal to its non-dominated level (1 if it belongs to the first front, 2 for the second front, and so on). In this order, minimization of rank is assumed. After ranking the solutions, a population of $n$ offsprings, $Q_0$, is created using recombination, mutation and a diversity-preserving binary tournament selection operator based on crowding distances. According to this selection operator, known as crowded tournament selection, a solution $i$ wins a tournament with another solution $j$ if he following conditions are true:

- If solution $i$ has a better (smaller) rank than solution $j$,
- If the ranks of both solutions are the same, the solution located in the less crowded region is preferred.

The NSGA-II procedure for the $i$ th generation is outlined in the following steps:

1. Let $Q_i$ be the offspring population, which is created from the parent population $P_i$ using the crowded tournament selection operator, recombination and mutation.
2. A combined population $R_i = P_i \cup Q_i$ of size $2n$ is formed.
3. $R_i$ is ordered according to non-domination (*i.e.*, each solution is assigned a rank). Since all previous and current population members are included in $R_i$, elitism is ensured. Solutions belonging to the best front, $F_1$, are the best solution in the combined population $R_i$.
4. If the size of $F_1$ is smaller than $n$, all members of the set $F_1$ are chosen for the new population $P_{i+1}$. The remaining members of the population $P_{i+1}$ are chosen from subsequent non-dominated fronts in the order of their ranking until no more sets can be accommodated. If $F_j$ is the last front from which individuals can be accommodated in the population, but not all the members can enter in the population, then a decision needs to be made to choose a subset of individuals from $F_j$. In order to decide which members of this front will win a place in the new population, NSGA-II uses once again the crowding distance strategy to favor solutions located in less crowded regions.

MOGP is a form of MOEA where individuals are encoded by means of rich structures (typically tree structures) (*Koza, 1992*). For instance, an individual can encode a computer

program. In this work, each individual encodes a tree-structured Boolean query. We refer the reader to *Deb et al. (2002)* and *Coello Coello, Lamont & Van Veldhuizen (2007)* for a detailed explanation of NSGA-II, MOGP, and MOEAs in general.

# RELATED WORK

## Evolutionary algorithms for information retrieval

Several techniques based on EAs have provided useful solutions to personalize information access (*Venturini, Carbó & Molina, 2008*; *Lin, Yeh & Liu, 2012*; *Zuo et al., 2015*; *Sadeghi & Asghari, 2017*; *Ibrahim & Landa-Silva, 2018*; *Mhawi et al., 2022*). EAs have also been applied for query refinement, which includes learning query terms, query-term weights and query operators (*Singh & Sharan, 2018*; *Sharma, Pamula & Chauhan, 2019*). A seminal approach that uses EAs to automatically evolve a population of Boolean queries is presented by *Smith & Smith (1997)*. In this work, the authors use the Cranfield Collection to generate an initial population of queries, which is then evolved by applying GP with an aggregative fitness function based on the classic precision and recall metrics. *Larsen et al. (2000)* propose to select terms based on a previous classification task made by a GA whose evolution depends on the user feedback opinions. *Venturini, Carbó & Molina (2008)* use a GA to infer users' preferences based on previous users' decisions. Another approach that addresses the problem of information retrieval using GP is presented in *Kulunchakov & Strijov (2017)* and proposes a solution to construct ranking models for information retrieval that depend on the document description. The main contribution is the definition of a new technique to avoid stagnation and to control the structural complexity of the generated models. The work presented by *Singh & Sharan (2018)* uses several single GAs to select the optimal candidate terms to expand the original query, and uses the expanded query to retrieve a final set of relevant documents. *Sharma, Pamula & Chauhan (2019)* use a hybrid evolutionary algorithm approach to automatically extract candidate keywords to expand the original query. The new keywords are selected based on the relevance of retrieved documents, whose ranking score is given by the Okapi BM25 ranking function. The Webnaut system (*Nick & Themis, 2001*) creates queries by combining keywords and logic operators generated by GAs. The best sets of keywords and logical operators are then used to construct high-dimensional concepts representing the user interests. *Chen & Shahabi (2002)* propose to improve the current user recommendation list by using a GA that starts from historic experts' data and incorporates the new users' navigation behaviors through the evolution process. The work presented by *Malo, Siitari & Sinha (2013)* shows the effectiveness of evolving queries based on Wikipedia concepts by means of a co-evolutionary GP scheme. The strategy starts from a set of relevant and irrelevant documents provided by the user and evolves the queries evaluating them in terms of F-measure. *Baeza-Yates et al. (2019)* propose a GP algorithm for ranking web documents called CombGenRank. This algorithm applies elitism to supporting web document ranking. The proposed algorithm is motivated by search services based on cloud computing. Another tool for topic-based recommendation is presented in *Eckroth & Schoen (2019)*, where the authors describe a method based on GA to select news stories about artificial intelligence for AAAI's weekly AI-Alert. *Mhawi et al. (2022)* propose a

document indexing method that uses an EA-based retrieval system. This EA-System learns user-relevant queries, allowing retrieving the most relevant documents to the user's interests.

Several proposals have demonstrated that Pareto-based multi-objective strategies offer a natural way to evolve populations of effective queries. *Cordon, Herrera-Viedma & Luque (2006)* combine the Strength Pareto Evolutionary Algorithm (SPEA) with GP to evolve a population of Boolean queries. Each query is codified as an expression tree where the inner nodes are Boolean operators and the terminal nodes are terms. In a subsequent work, *López-Herrera, Herrera-Viedma & Herrera (2009)* compare the performance of four well-known MOEAs (MOGA, SPEA, SPEA2 and NSGA-II) in the problem of automatically learning queries by IQBE. A GP-based algorithm was designed for each of the evaluated MOEA schemes, which allowed the authors to conclude that the NSGA-II scheme outperforms the others.

Our proposal shares insights and motivations with the studies presented in *Cecchini et al. (2008*, *2010*, *2018)*. In the first work (*Cecchini et al., 2008*) the authors show the effectiveness of EAs to automatically learn topical queries to retrieve material from the web guided by the user's thematic context. In particular, the work offers a mechanism for assessing query performance in the absence of relevance judgments. It also provides an analysis of the impact of different mutation rates on performance and proposes a technique to augment queries with new vocabulary extracted from the relevant documents that are incrementally retrieved. The queries, the user context and the retrieved results are treated as vectors of terms in order to calculate similarity measures, such as cosine similarity. The evaluation is based on the quality of the best queries at each generation of the GA, and the performance improvement is measured as the increase in the quality value as the generations pass.

The work presented by *Cecchini et al. (2010)* is related to our proposal in addressing the problem of topic-based search using a MOEA to account for multiple objectives. It also uses a topic description as a source for query generation and a topic ontology to train and test the framework. The authors treat queries as sequences of terms following the disjunctive semantics, which means that a single matching term is sufficient to retrieve a document. The objectives to be maximized are Precision at rank 10 and Recall and three strategies are compared: single objective EAs, a Pareto-based MOEA, and an aggregative MOEA. For each strategy the authors evaluate whether the queries at the initial generations are outperformed by those in subsequent generations, and then evaluate the evolved queries on a test set. The study also reports the results obtained by two baseline methods that are based on supervised and non-supervised generalizations of the Rocchio's method (*Amati & Van Rijsbergen, 2002*). The authors discuss the limitations of the single-objective EA strategy and show that the Pareto-based and aggregative MOEAs have comparable performance.

In *Cecchini et al. (2018)* the authors propose a family of strategies for evolving topical queries that account for the problem of achieving high recall at the population level. In order to favor diversity, the authors propose new fitness functions, referred to as retrospective fitness functions, that penalize queries that retrieve the same results as other

queries in the same population. The proposed strategies assume a total order over the queries, and the retrospective fitness functions of a query are calculated taking into account the results of the queries that precede it. The authors also show that the use of multiple populations favors diversity at the genotype level.

The present study differs from the previously mentioned ones in several respects: (1) we incorporate the use of Boolean operators to build the queries, which provides a more expressive syntax; (2) we propose the use of GP as the underlying evolutionary technique, which allows to encode the queries' syntax trees; (3) we define new fitness functions specifically designed to attain high global recall and diversity; and (4) we evaluate seven different fitness function combinations based on classical GP settings.

## Diversity maximization and diversity preservation in EAs

The problem of diversity maximization is concerned with selecting from a population a subset of items that are as diverse as possible. This problem is commonly known as the Maximum Diversity Problem and it is known to be NP-hard (*Kuo, Glover & Dhir, 1993*). Due to its intractability a number of heuristics have been proposed to obtain approximate solutions for this problem. Besides its time complexity, another challenge associated with this problem is the definition of a diversity measure on subsets of selected elements. A widely adopted measure is the Max-Sum Diversity measure (*Ghosh, 1996*), which is used to maximize the sum of distances among the selected elements. Alternatively, the Max-Min Diversity measure (*Resende et al., 2010*) is used to maximize the minimum distance among the selected elements. A natural approach to deal with this problem is by using evolutionary algorithms (EAs) that attempt to maximize the diversity among a selected subset of items (*Katayama & Narihisa, 2005*). *Vera et al. (2017)* proposed a Multi-Objective Evolutionary Algorithm (MOEA) to simultaneously optimize several diversity measures.

Closely related to the problem of diversity maximization is the problem of diversity preservation in EAs. The latter problem is typically associated with the problem of optimizing one or many objective functions while preserving diversity among individuals. Diversity can be based on either the individuals' genotype representation (*i.e.*, computed in the decision variable space), phenotype representation (*i.e.*, computed in the expression trait space), or phenotype evaluation (*i.e.*, computed in the objective function space). Our problem domain is suitable for computing similarity among individuals by any of these approach. For example, an approach based on genotype representation similarity could favor diversity by looking at the representations of the individual queries, computing the similarity among them and penalizing those queries that are similar to many other queries. On the other hand, an approach based on phenotype representation similarity could take into consideration information associated with the retrieval results of each query. A simple definition of phenotype similarity between queries could be based on counting the number of common results returned by the queries. Finally, the phenotype evaluation similarity could be assessed by computing a spread metric on the Pareto front.

In order to address issues related to diversity preservation several Niching methods have been developed. The best-known strategies aimed at niching are crowding (*Mengshoel &*

*Goldberg, 2008*) and fitness sharing (*Goldberg & Richardson, 1987*). In both cases, the goal is to obtain solutions uniformly distributed over the Pareto Front. Crowding methods favor individuals from less crowded regions. On the other hand, fitness sharing is a mechanism for preserving diversity that modifies the landscape by reducing the payoff in densely populated regions. The main disadvantage of the fitness sharing method is that it is necessary to provide a niche size parameter, which defines a neighborhood of solutions in the objective function space.

The computational cost of estimating diversity among individuals is typically very high. A less expensive technique that favors the exploration of diverse solutions is local selection (*Menczer, Degeratu & Street, 2000*), an approach that is similar to, but more efficient than fitness sharing. It has the advantage of allowing parallel implementation for distributed tasks, an important pro in the information retrieval scenario. This is achieved by evaluating the fitness function by an external environment that provides appropriate data structures for maintaining shared resources. This technique is amenable to the information retrieval task, where each retrieved document can be marked so that the same document does not yield payoff multiple times.

Population fitness measures have been proposed to evaluate the diversity of a population as a whole. The concept of population fitness is present in biology (*Markert et al., 2010*) and is supported by the principle of community heritability (*Shuster et al., 2006*). In EAs, an example of population fitness known as global fitness is incorporated in the Parisian approach (*Collet et al., 2000*). In this algorithm, a global solution is built by the combination of the information provided by several members of the population, whose individual performances are evaluated by a *local fitness* function.

The algorithm proposed in *Zhou et al. (2022)*, uses a diversity maintenance mechanism based on the global and local diversities by considering them collaboratively. The similarity between two individuals is defined as the cosine similarity between their objective vectors. Then, a balance strategy is used to adjust weights of convergence for global diversity and local diversity according to the population. Each solution reaches a score that represents its performance in diversity and convergence simultaneously.

Another strategy that favors diversity preservation is based on the concept of multiple populations (*Gupta & Ghafir, 2012*). The idea behind multi-population EAs consists in defining subpopulations which evolve independently; thus the unique features of each subpopulation can be effectively preserved, and the diversity of the entire population is benefited. In some algorithms, these subpopulations can exchange information, as it is the case in Multi-population Coevolutionary Algorithms (*Shi et al., 2014*), but in other cases a complete isolated evolution is executed over each subpopulation (*Cochran, Horng & Fowler, 2003*).

MOEAs deteriorate their search ability when solving many-objective optimization problems (MaOPs), *i.e.*, problems with more than three objectives. *Li et al. (2019)* propose a dynamic convergence-diversity guided evolutionary algorithm for MaOPs by employing the decomposition technique where a set of uniformly distributed reference vectors is used to divide the objective space into K subspaces. Each subspace has its own subpopulation and evolves simultaneously with the others.

## Diversity in information retrieval

The problem of diversification of search results has been extensively studied by the information retrieval community. Clustering offers an approach to maximizing diversity in information retrieval since a form of diversity can be attained by selecting items from different clusters (*Yu et al., 2018*). Another diversity maximization approach is based on nonnegative matrix factorization, which applies network dimensionality reduction and estimates relevance in a low dimension space (*Meng et al., 2018*). Other approaches attempt to identify diverse results for a query by accounting for different "aspects" associated with a query (*Clarke et al., 2008*; *Agrawal et al., 2009*; *Bouchoucha, He & Nie, 2013*). The approaches mentioned primarily concentrate on identifying various results linked to a particular query. In contrast, the methods presented in this article focus on creating diverse queries associated with a subject of interest. *Carpineto, D'Amico & Romano (2012)* take the notion of subtopic as a basis to optimize the relevance of results at the document set level by considering clustering and diversification of search results. In *Ghosal et al. (2021)*, the authors explore the use of textual entailment models to identify document novelty at a semantic level. The approach is intended to automatically detect novel information in a given text by combining machine learning and natural language processing. *Karakaya & Aytekin (2018)* present a method to diversify the top-N items on a recommendation list. Their proposal tries to give the user a global view of possible user's interests, rather than only suggesting similar items. The method reorganizes the results by looking for sub-topics. The performance is evaluated based on precision, recall, and user behavior, and showed that the user's overall likeness goes beyond classical factors.

While the above-mentioned methods attempt to diversify search results, other methods aim at diversifying queries. This can be done by automatically generating diverse but semantically related queries. Some of these strategies rely on the use of query-URL clickthrough data (*Ma, Lyu & King, 2010*) while others are based on external semantic sources, such as Wikipedia (*Hu et al., 2013*; *Baggio et al., 2019*) or Wordnet (*Zheng et al., 2014*). In *Baggio et al. (2019)* we analyzed whether the use of Wikipedia concepts instead of terms positively affects the diversity of the search results. However, improvements were only obtained in terms of interpretability and readability of the queries for the user, not in terms of the results' diversity. *Patro, Niaz & Prasath (2023)* propose an information retrieval framework that identifies different aspects of information and organizes clusters based on similarity to each aspect. In this way, the final retrieved results cover diverse information needs. Before the final response, the framework applies query expansion to refine the final set of clusters. *Kim & Croft (2014)* attain query diversification by taking a document from which multiple aspects (subtopics) can be identified. Query diversification methods have also been applied to develop query suggestion or auto-completion mechanisms (*Kharitonov et al., 2013*). *Li, Schijvenaars & de Rijke (2017)* investigate empty-result queries and discuss how to avoid them. A review of a large number of methods proposed in the literature for search result diversification is presented in *Santos, Macdonald & Ounis (2015)*.

# A MOGP FRAMEWORK FOR SEARCHING TOPIC-RELEVANT AND DIVERSE RESOURCES

This work presents a framework for topic-based search that incorporates precision, global recall, and diversity as key objectives to guide the search process. The proposed framework takes a topic description and a collection of labeled documents to characterize the user information needs. It then applies MOGP to evolve a population of topical Boolean queries. In the analysis reported in this article we use the topic description and labeled documents from the Open Directory Project (https://dmoztools.net/) (ODP). However, other collections can be used as long as a description of the topic of interest is available and the documents in the collections are classified as relevant or irrelevant to the given topic.

The proposed framework generalizes the works presented by *Cecchini et al. (2010)* in a number of fundamental ways. First, it applies GP instead of GA and hence learns tree-structured Boolean queries instead of unstructured queries. Also, besides attempting to attain high precision and recall for each individual query it considers the collective performance of the entire population of queries with the purpose of achieving high global recall and diversity. While the problem of diversity preservation was also addressed in *Cecchini et al. (2018)*, the new framework introduces novel entropy-based fitness functions as well as a fitness function based on the Jaccard similarity index to measure the similarity between sets of results. Next, we describe the components of the proposed MOGP approach.

**Representation of individuals** The decision variable space $S_q$ contains all valid Boolean queries that can be formulated to a search interface and each population is considered to be a multiset of queries (note that the same query may appear more than once in the same population). In GP, individuals are commonly represented by means of tree structures. In this work, the Boolean queries are encoded as expression trees, whose terminal nodes are query terms and whose inner nodes are the Boolean operators AND, OR, or NOT. The starting population comprises $n$ Boolean queries. Each query in the initial population is a tree-structured individual where the internal nodes are randomly chosen from the set of Boolean operators {AND, OR, NOT} and the leaf nodes are randomly chosen from the set of terms of the description of the given topic (available from ODP). The maximum depth of each individual is 17. The queries are internally stored in preorder. For example:

$q_1$: NOT(OR(`comets´, `sun´), AND(`nine´, `asteroids´))

$q_2$: OR(`planets´, AND(NOT(`hydrogen´, `collections´), NOT(`radioactively-labeled´, `collections´)))

The tree representation for $q_1$ and $q_2$ can be found in Fig. 1. Note that the NOT operator used by Lucene is binary and therefore the query NOT(A,B) should be interpreted as $A \wedge \neg B$. Thus, the queries given in the previous example should be interpreted as:

$q_1$: (`comets´ $\vee$ `sun´) $\wedge \neg$ (`nine´ $\wedge$ `asteroids´)

$q_2$: (`planets´ $\vee$ ((`hydrogen´ $\wedge \neg$ `collections´) $\wedge$ (`radioactively-labeled´ $\wedge \neg$ `collections´)))

The sizes of the initial queries are bounded by a predefined constant. However, the sizes of some queries in subsequent generations can exceed this limit as a result of applying the

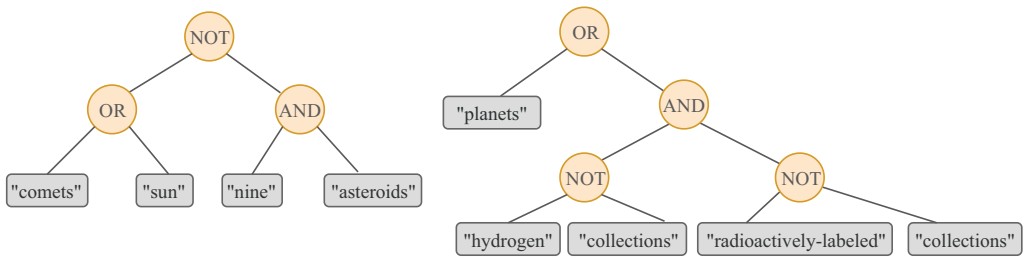

**Figure 1 A graphical representation of a GP individual.** A query based on terms extracted from the description of the topic *Solar System* (left) and an evolved query that incorporates novel terms learned from the retrieved resources (right).

crossover operator. The initial population of queries is comprised solely of terms extracted from the initial topic description. Through the process of mating, these queries are consistently recombined in novel ways, resulting in the creation of fresh solutions. In particular, the mutation mechanism is implemented in such a way that novel terms, *i.e.*, terms that are not in the initial topic description, are brought into play. Later in this section, we will elaborate on how the mutation pool provides new terms. Gradually, the most efficient queries will gain greater prominence.

**Genetic operators and evolution** A set of operators that pick (select), merge (recombine), and modify (mutate) queries from the current population determines a fresh generation of queries. Once the initial population is obtained, each individual is assigned a fitness value and the NSGA-II method is applied (*Deb et al., 2002*).

While some of the queries in the new population remain unchanged, others are recombined or mutated. The recombination (or crossover) consists in picking up two individuals from the population as parents, randomly selecting a crossover point in each individual, and exchanging their subtrees starting at that point. As a result, a pair of offspring is obtained from the two parents. This process is illustrated in Fig. 2. Parents and offspring ($2n$ individuals) compete to conform the next generation of queries of size $n$. The crossover operator employed in our proposal is referred to as the single-point technique. The selection of every pair of parents that are combined by the Crossover operator is carried out using the Double Tournament selection method with a tournament size of 10.

Mutation allows to introduce small changes on some individuals of the population. These changes consist in replacing a randomly chosen leaf of the individual (a query term $t_i$) by another term $t_j$ obtained from a mutation pool. The tree-based GP mutation is illustrated in Fig. 3. The mutation pool is a set[1] that is initially comprised of terms extracted from the topic description. As the system collects relevant documents over successive generations, new terms are extracted and added to the pool. The pool has a 10.000 terms limit and does not contain stop-words. If the capacity is exceeded at any point of the evolution, random terms are removed to make space and let those who fell out of the limit be incorporated into the pool. In this manner, the introduction of new terms enables a wider exploration of the search space.

At first, both the mutation and crossover rates were fixed based on an extensive preliminary analysis reported in *Cecchini et al. (2008)*. These two values were initially set as

[1] No repetitions are allowed

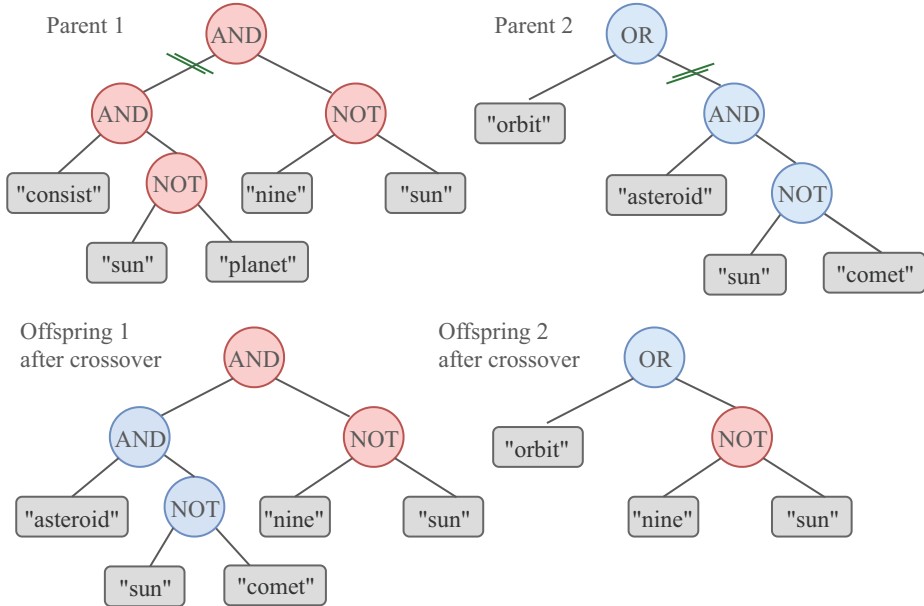

**Figure 2 A graphical representation of the GP crossover.** To produce two new offspring, a one-point crossover involves the exchange of a subtree between the two parents. The crossover probability is set to $cxPb = 0.7$.

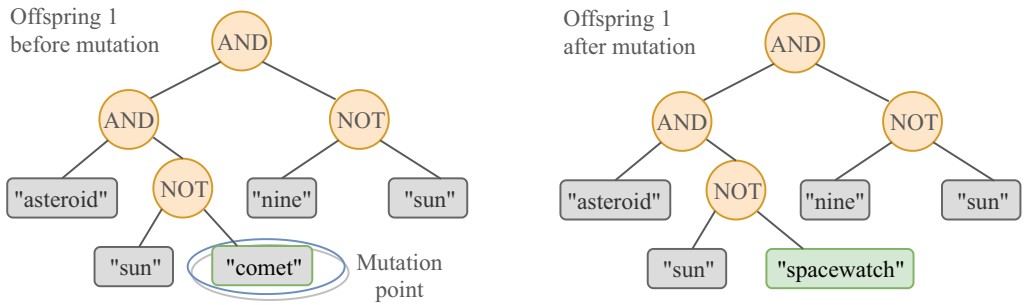

**Figure 3 A graphical representation of the GP mutation.** Mutation is applied at the term level, meaning that only a term (leaf) randomly selected from the tree is replaced with probability $mutPb = 0.3$ by a term randomly selected from the mutation pool.

$mutPb = 0.03$ and $cxPb = 0.7$, respectively. The reason for giving mutation such a limited role is the generally shared view that crossover has a large shuffling effect, acting in some sense as a macromutation operator (*Eiben & Smith, 2015*). However, considering that topic descriptions only account for a limited number of terms to build the first generation of queries and given the goal of attaining high diversity, it becomes necessary to incorporate new genetic material to allow a broader exploration of the search space. A variety of mutation rates that go from $mutPb = 0.2$ (*Cordon, Herrera-Viedma & Luque, 2006*) to $mutPb = 0.9$ (*Malo, Siitari & Sinha, 2013*) have been analyzed in the literature. After substantial experimentation, we decided to maintain the classical crossover probability $cxPb = 0.7$ and to use a mutation rate $mutPb = 0.3$.

In our multi-objective scenario, the actual selection is carried out by using the NSGA-II method that provides the functionality of probabilistically selecting the highest quality queries from the current population to produce the next generation of queries. The evolution of the Boolean queries is guided by objectives defined both at the individual and at the population level. As explained next, the goal is to attain high precision for every single query and high global recall (hence diversity) for the collection of evolved queries.

**Fitness functions** The fitness functions define the criteria for assessing the quality of a query. Assuming that it is possible to distinguish relevant documents from non-relevant ones, several classical (*Manning, Raghavan & Schütze, 2008*) and *ad-hoc* measures of query effectiveness can be adopted as fitness functions.

Let $Rel_t$ be the set containing all the documents that are relevant to topic $t$ ($Rel_t$ can be obtained from a topic ontology such as ODP, as explained in the next section), and let $Ret_q$ be the answer set returned by a search engine when $q$ is used as a query. It is worth mentioning that the retrieved results are ranked based on the default Scorer function provided by Lucene's implementation. Lucene scoring is based on a combination of the Vector Space Model of Information Retrieval and the Boolean model to determine how relevant a given document is to a query. We will assess precision by only considering the top-10 retrieved results (ranked by Lucene scoring) rather than the entire answer set. We define $Ret@10_q$ as the set of top-10 ranked documents in $Ret_q$ (note that if the search engine returns less than 10 results for the query $q$, the size of $Ret@10_q$ might be less than 10). The measurement of retrieval accuracy in our approach employs precision at rank 10 as a fitness function. The metric precision at rank 10 for a query $q$ and a topic $t$ can be defined as:

$$Precision@10(q, t) = \frac{|Rel_t \cap Ret@10_q|}{|Ret@10_q|}.$$

On the other hand, given a query $q$ for a topic $t$, the recall of $q$ is defined as the fraction of relevant documents associated with that topic that are retrieved:

$$Recall(q, t) = \frac{|Rel_t \cap Ret_q|}{|Rel_t|}.$$

We define a new fitness function inspired by the information theoretic notion of entropy, to which we refer to as entropic recall. The rationale behind the entropic recall measure is that if a query tends to retrieve the same resources as other queries in the population P, it will not contribute with novel material so its fitness value should be downweighted. This way, this measure penalizes those queries that do not help to achieve diversity while it favors those that can retrieve resources that other queries are unable to recover. This results in the entropic recall fitness function's output for a given individual, $q_i$, being influenced not only by its own characteristics but also by the resources retrieved by other individuals in the population. Our definition of the entropic recall fitness function is as follows:

$$Entropic - Recall(q, t, P) = \frac{\sum_{d_i \in Rel_t \cap Ret_q} \frac{IQF(d_i, P)}{log(|P|+1)}}{|Rel_t|},$$

Here, $IQF(d_i, P) = \log((|P| + 1)/n_i)$ and denotes the inverse query frequency of document $d_i$ across the collection of documents retrieved by all queries in population $P$, where $n_i$ is the number of queries retrieving $d_i$.

Note that all documents $d_i$ considered in this formula are such that $d_i \in Rel_t \cap Ret_q$ and hence will always be retrieved by at least one query. As a consequence, $n_i$ will never be equal to 0. On the other hand, to prevent the problem of taking the logarithm of zero, 1 is added to the population size. In terms of numerical value, $IQF(d_i, P)$ can be viewed as a logarithmic approximation of the inverse probability that a randomly chosen query $q$ from a population of size $|P| + 1$ would retrieve document $d_i$. $IQF(d_i, P)$ is divided by $log(|P| + 1)$ to ensure that its value is in the interval $[0, 1]$. As a result, $IQF(d_i, P)/log(|P| + 1)$ will return a value close to 1 if queries in the population $P$ rarely retrieve document $d_i$, while it will return a value close to 0 if most queries in $P$ retrieve $d_i$. By calculating the sum of the inverse query frequency ($IQF$) of all relevant documents retrieved by query $q$, the entropic recall metric evaluates the degree of uniqueness associated with the query.

As part of our contribution, we define entropic precision at rank 10 based on precision at rank 10, but similar to entropic recall we favor queries that retrieve documents that are rarely retrieved among the top-10 results by other queries. Our definition of *Entropic-Precision@10* is as follows:

$$Entropic - Precision@10(q, t, P) = \frac{\sum_{d_i \in Rel_t \cap Ret@10_q} \frac{IQF@10(d_i, P)}{log(|P|+1)}}{|Ret@10_q|},$$

where $IQF@10(d_i, P) = \log((|P| + 1)/n@10_i)$ and it represents the inverse query frequency of document $d_i$ across the collection of documents returned by all the queries in population $P$, where $n@10_i$ is the number of queries among the top-10 results that retrieve $d_i$.

Finally, we introduce another fitness function based on the classical Jaccard similarity index. This measure is defined specifically to favor the diversity of relevant results retrieved by a query with respect to the relevant results returned by other queries in the population. The idea is to measure how similar the answer set of relevant results returned by a query is to the answer set of relevant results returned by the rest of the queries in the population.

Let $Ret_{q_i}^{\star} = Ret_{q_i} \cap Rel_t$. Then, we compute *Jaccard—Similarity* as follows:

$$Jaccard—Similarity(q_i, t, P) = \frac{\sum_{q_j \in P, j \neq i} J - Sim - Index(Ret_{q_i}^{\star}, Ret_{q_j}^{\star})}{|P| - 1},$$

where $J - Sim - Index(Ret_{q_i}^{\star}, Ret_{q_j}^{\star}) = |Ret_{q_i}^{\star} \cap Ret_{q_j}^{\star}|/|Ret_{q_i}^{\star} \cup Ret_{q_j}^{\star}|$. High values of *Jaccard—Similarity* reflect diversity loss and hence we attempt to minimize this measure.

## EVALUATION

### Data

To run our evaluations, the data were collected as previously described in *Cecchini et al. (2010)* using the evaluation framework introduced in *Cecchini et al. (2011)*. To ensure the quality of our *corpus*, certain restrictions were placed on the selection process (for the dataset description and download (https://data.mendeley.com/datasets/9mpgz8z257/1), please refer to *Lorenzetti, Maguitman & Baggio (2019)*. The *corpus* comprises over 350,000 webpages classified into 448 topics. To index these webpages and run our experiments we used the Python extension for the Lucene framework (http://lucene.apache.org/pylucene/). All terms underwent Porter stemming and stopwords were eliminated. The topics were divided into two parts, with two-thirds of the webpages used to create the training index, while the remaining one-third was allocated for building the testing index. The evaluations reported in this article were carried out using the topic descriptions extracted from 25 randomly chosen topics. Each topic has its own ODP description that typically consists of a few sentences briefly describing the content of the webpages indexed under that category.

### Performance measures

The performance of a strategy can be measured by the collective effectiveness of the population of queries evolved by the strategy. For a given topic $t$, the following metrics provide a means to evaluate the effectiveness of a query population $P$:

- $\overline{Precision@10}$. The mean precision at rank 10 is calculated as the average of $Precision@10$ scores obtained for all queries within population $P$. For a given population $P$ and a topic $t$, this metric is computed using the following formula:

$$\overline{Precision@10}(P, t) = \frac{\sum_{q \in P} Precision@10(q, t)}{|P|}.$$

- $Global - Recall$. The global recall metric evaluates the performance of the entire query population in achieving high recall scores. The union of results returned by a query population $P$, denoted by $A(P)$, is defined as the collective set of results returned by all queries in $P$: $A(P) = \bigcup_{q_i \in P} Ret_{q_i}$, where $Ret_{q_i}$ is defined in the usual way as the set of results returned by $q_i$. To calculate $Global - Recall$ for a given population $P$ and topic $t$, we use the following formula:

$$Global - Recall(P, t) = \frac{|A(P) \cap Rel_t|}{|Rel_t|}.$$

- $\overline{Jaccard - Similarity}$. The mean Jaccard similarity metric results from averaging the values of $J - Sim - Index$ across all pairs of answer sets for queries in $P$ restricted to relevant documents. It allows to assess the diversity of results returned by a population of queries. As mentioned before, let $Ret_{q_i}^\star = Ret_{q_i} \cap Rel_t$. Then, we define $\overline{Jaccard - Similarity}$ as follows:

$$\overline{Jaccard - Similarity}(P, t) = \frac{\sum_{q_i, q_j \in P, i \neq j} J - Sim - Index(Ret^\star_{q_i}, Ret^\star_{q_j})}{|P| \cdot (|P| - 1)},$$

where $J - Sim - Index(Ret^\star_{q_i}, Ret^\star_{q_j}) = |Ret^\star_{q_i} \cap Ret^\star_{q_j}| / |Ret^\star_{q_i} \cup Ret^\star_{q_j}|$.

## MOGP strategies

To evaluate the proposed fitness functions, we experimented with seven MOGP strategies based on various combinations:

- **Co1**: *Precision*@10 and *Recall*. This is a baseline strategy that attempts to maximize the relevance of the top 10 retrieved results (*Precision*@10) and the proportion of relevant documents that are successfully retrieved from the entire collection (*Recall*) Balancing *Precision*@10 and *Recall* is a challenge, as there is often a trade-off between the two.

- **Co2**: *Precision*@10 and *Entropic − Recall*. This strategy combines the classical *Precision*@10 metric with the novel *Entropic − Recall*. As in the case of **Co1**, the goal is to maximize the relevance of the 10 retrieved results and the proportion of relevant documents that are successfully retrieved from the entire collection. Different from *Recall*, *Entropic − Recall* favors queries that retrieve results that other queries are unable to recover.

- **Co3**: *Entropic − Precision*@10 and *Entropic − Recall*. This strategy is similar to **Co1** in that it attempts to simultaneously maximize the relevance of the top 10 retrieved results and the ratio of relevant documents effectively retrieved from the entire collection. However, it integrates both objectives with the goal of retrieving results that other queries fail to capture.

- **Co4**: *Precision*@10, *Recall* and *Jaccard − Similarity*. This strategy complements **Co1** by incorporating the objective of minimizing the similarity of the set of results retrieved by the query and the set of results retrieved by each of the other queries. This is carried out by minimizing the *Jaccard − Similarity* fitness function.

- **Co5**: *Precision*@10, *Entropic − Recall* and *Jaccard − Similarity*. This strategy is similar to **Co4**, except that it replaces the objective *Recall* by *Entropic − Recall*, giving preference to queries that retrieve results not recovered by the other queries.

- **Co6**: *Precision*@10 and *Jaccard − Similarity*. This strategy attempts to maximize the relevance of the top 10 retrieved results, while also minimizing the similarity of the result set retrieved by the query and those obtained from each of the other queries. It does not incorporate the *Recall* metric or any of its derivatives.

- **Co7**: *Precision*@10, *Jaccard − Similarity* and $|Ret^\star|$. This strategy augments **Co6** by also attempting to maximize the size of the retrieved set of relevant documents $Ret^\star$. Although it does not incorporate the *Recall* metric or any of its variations, we expect to attain high coverage through the objective of maximizing $|Ret^\star|$.

**Table 1  GP parameters and operators used for the seven combinations of objectives (Co1 to Co7).**

| GP configuration | |
|---|---|
| Number of runs | 5 |
| Number of topics | 25 |
| Number of generations | 150 |
| Number of subpopulations | 1 |
| Population size | 100 |
| Crossover probability, $cxPb$ | 0.7 |
| Mutation probability, $mutPb$ | 0.3 |
| Initial tree depth | 1–5 |
| Maximum crossover depth | 17 |
| Parent selection operator | selDoubleTournament |

In all the above combinations the objectives are maximized, except for the case of $Jaccard - Similarity$, which is minimized. For comparison purposes, combination **Co1** was taken as a baseline, which is based on classical measures of $Precision@10$ and $Recall$.

The strategies were first assessed through a case study. Then a comparison in terms of $\overline{Precision@10}$, $Global - Recall$, and $\overline{Jaccard - Similarity}$ was completed on the training set by running each strategy 5 times on each of the 25 selected topics. Finally, the queries evolved by the strategies were compared on the testing set by means of the proposed performance metrics.

Table 1 summarizes the parameter configuration used during the training stage. These parameters were selected based on previous results and guidelines found in the GP literature (*Cordon, Herrera-Viedma & Luque, 2006*; *Malo, Siitari & Sinha, 2013*; *Eiben & Smith, 2015*) and our own previous work (*Cecchini et al., 2008*, *2010*, *2011*, *2018*). Instead of focusing on the analysis of various parameter configurations, which can become complex due to their combinatorial nature, we prioritized the exploration of the proposed MOGP strategies. With this objective in mind, we evolved queries employing the MOGP strategies, resulting in the generation of more than 22.300.000 queries.

A common problem in GP is that queries experience a significant increase in size throughout the generations, therefore we selected the potential parents to perform the crossover and mutation using the selDoubleTournament operator, an alternative selection operator also used as a bloat control technique (*Luke & Panait, 2002*). This operator uses a special sample function to select candidates, which is another tournament based on the size of the individuals. The actual selection process is then carried out based on fitness. Query depth is limited to 17, which is the standard maximal depth limit for tree-based GP to prevent excessive growth.

For each topic description, the training index was used to evolve queries using five independent runs. For each run we considered 150 generations with a population size of 100 queries. The test set was used to assess the effectiveness of the evolved queries on a new *corpus* for a specific topic. It's worth noting that while the train and test sets comprise the

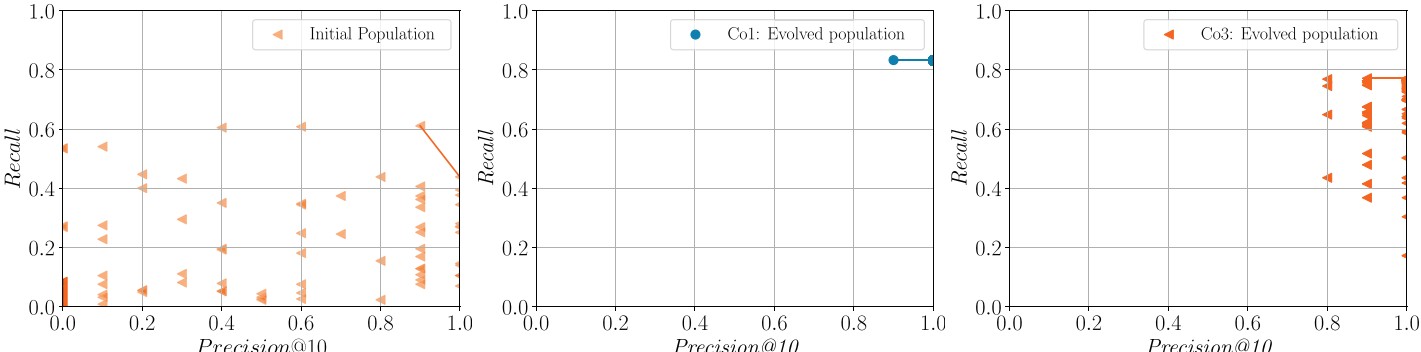

**Figure 4 Sample populations of queries obtained for the topic, Solar System.** The initial population (identical for all the strategies) is shown on the left. The other charts correspond to the evolved populations for the baseline **Co1** (center) and **Co3** (right). The convex hulls of the Pareto fronts are represented by lines. The dispersion observed in the three charts allows to appreciate the diversity at the phenotype evaluation level of the initial population (random queries), a population evolved by a strategy that does not prioritize diversity (**Co1**), and a population evolved by a strategy that attempts to preserve diversity **Co3**.

same topics (and hence the same topic descriptions), they contain different documents (webpages).

As a first step in the performance analysis, we looked at the evolution of the queries associated with the individual topics. For illustrative purposes, we report the analysis on the topic Solar System. The description available from the ODP distribution for this topic is *"The Solar System currently consists of the Sun, nine planets, and the many asteroids and comets that also orbit the Sun."* The initial queries were generated by randomly selecting terms from this description. New vocabulary was incrementally learned based on the retrieved material that was relevant to the topic. Some of the learned terms, such as "mars", "saturn" and "astrobiology", were used to automatically build subsequent queries helping to retrieve novel but topic-related content.

Figure 4 shows the charts corresponding to the *Precision@10* and *Recall* values of the queries evaluated on the topic Solar System based on a single run. For illustrative purposes, we only show the charts for the initial generation and the last generations of **Co1** (taken as a baseline for comparison) and **Co3** (taken as an example of strategy that incorporates a diversity preservation mechanism). The initial population is randomly generated and, as a consequence, it is highly diverse. However, as expected, the performance of these random queries is low in general, both in terms of *Precision@10* and *Recall*. On the other extreme, the queries evolved by strategy **Co1** have high *Precision@10* and *Recall*, but there is an evident loss in phenotypic representation diversity. Note that the strategy that incorporates a mechanism for diversity preservation (**Co3**) preserves phenotypic representation diversity. Also, individuals in the Pareto front achieve *Precision@10* and *Recall* comparable to **Co1**.

## Evaluation of the MOGP strategies on the training set

A more rigorous analysis can be carried out by averaging the values of each performance metric over the five completed runs and the 25 analyzed topics. *Precision@10*,

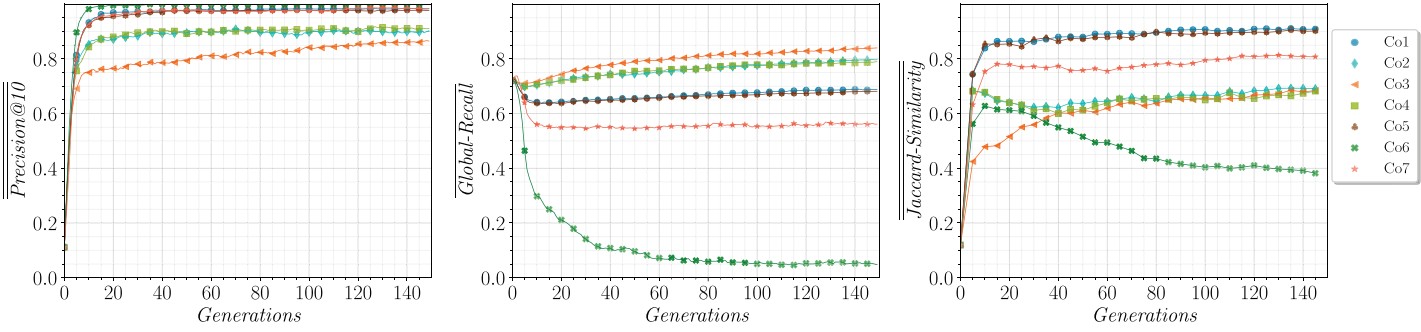

**Figure 5** Evolution of $\overline{Precision@10}$ (left), $\overline{Global - Recall}$ (center) and $\overline{Jaccard - Similarity}$ (right) on the training set (averaged over five runs on 25 topics).

$\overline{Global - Recall}$ and $\overline{Jaccard - Similarity}$ stand for the arithmetic mean of $\overline{Precision@10}$, $Global - Recall$ and $Jaccard - Similarity$, respectively, computed across all the runs and topics. The evolution of these metrics during training for the analyzed strategies is illustrated in Fig. 5. It can be observed that **Co3** achieves the highest $\overline{Global - Recall}$, followed by **Co2**, both of which are strategies with only two objectives, one of which is $Entropic - Recall$. Furthermore, it is interesting to note that when **Co1** is extended to formulate **Co4** by introducing $Jaccard - Similarity$ as a third objective, there is a substantial improvement in $\overline{Global - Recall}$, and it becomes comparable to **Co2** and **Co3**.

Note that **Co3** is the only strategy that does not attempt to maximize $Precision@10$ directly, but rather focuses on a reformulation of it, $Entropic - Precision@10$. However, it is noteworthy that it reaches an $\overline{Precision@10}$ value close to 0.9, which represents an excellent performance. In addition, **Co3** is highly effective in terms of $\overline{Jaccard - Similarity}$ and has the highest $\overline{Global - Recall}$ among all the strategies. In general, we can observe that the strategies that incorporate $Entropic - Recall$ are more effective than those that use $Jaccard - Similarity$.

Strategy **Co6** has only two objectives and it has been incorporated for completeness. Although it achieves the best $\overline{Jaccard - Similarity}$, it is the worst in terms of $\overline{Global - Recall}$. This can be attributed to the fact that **Co6** has $Jaccard - Similarity$ as an objective function, thus the material recovered might be diverse but consists of a reduced number of documents. Note that **Co7** is a reformulation of **Co6** that incorporates the number of relevant retrieved results ($|Ret^\star|$) as a third objective to be maximized. This improvement notably enhances $\overline{Global - Recall}$ in comparison to the **Co6** strategy. However, combination **Co7** performs worse than **Co2**, **Co3**, and **Co4** in terms of $\overline{Jaccard - Similarity}$. Consequently, **Co3** is the preferred choice among them.

It is worth mentioning that some strategies attain similar performance, as is the case of **Co1** and **Co5**. For those cases, the computational time can be used as a criterion to prefer a strategy over another. Hence, **Co1** would be preferred over **Co5** because it involves only two objectives (instead of three) and it does not require the evaluation of global fitness

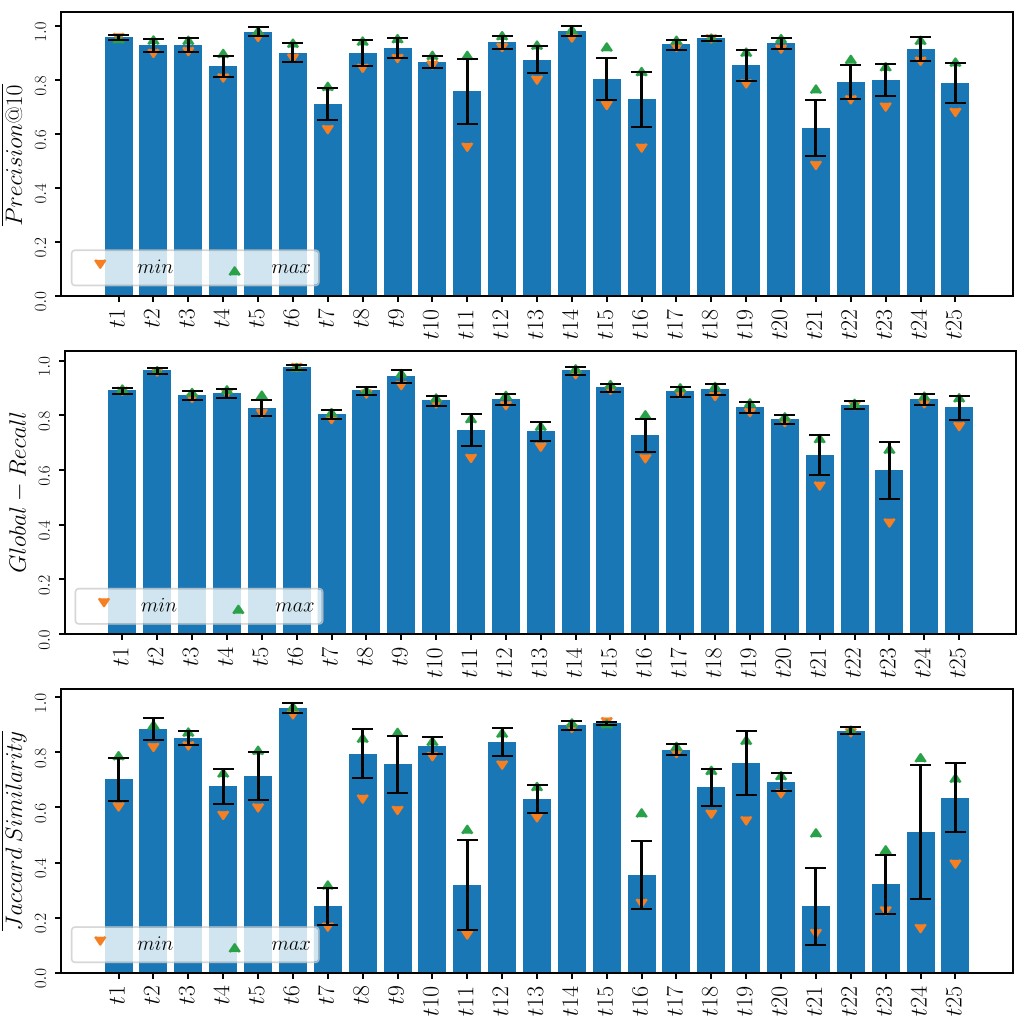

**Figure 6 Means and confidence intervals for $\overline{Precision@10}$, $\overline{Global - Recall}$ and $\overline{Jaccard - Similarity}$ at the 95% level of the Co3 strategy for each of the 25 topics across the five runs.** The maximum and minimum values obtained from the 5 runs are indicated using upward and downward-pointing triangles, respectively.

functions (*Entropic − Recall* and *Jaccard − Similarity*), which are more expensive to compute than the local fitness functions.

Finally, for a more comprehensive analysis and to illustrate the variability of one of the most effective MOGP strategies examined across different topics, we present in Fig. 6 the means and confidence intervals for **Co3** on each topic over the five runs. The small confidence intervals observed, especially for $\overline{Precision@10}$ and *Global − Recall*, serve as indicators of the method's stability and robustness.

## Evaluation of the MOGP strategies on the testing set

In this section, we evaluate the $\overline{Precision@10}$, $\overline{Global - Recall}$ and $\overline{Jaccard - Similarity}$ metrics achieved by the proposed MOGP strategies on the testing set. The questions addressed in the first place are: *How effective are the queries evolved by each of the proposed strategies in comparison to the initial queries that are formulated directly from the*

 

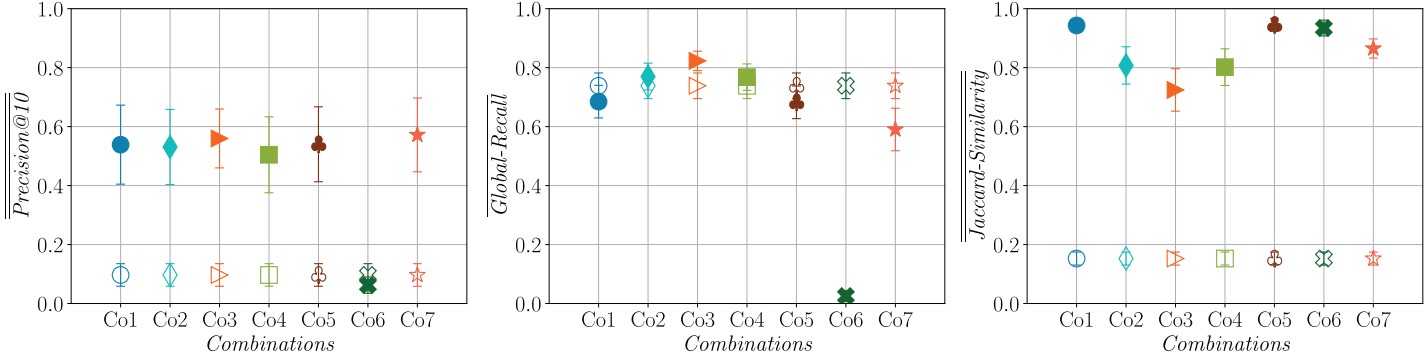

**Figure 7** Confidence intervals at the 95% level for $\overline{Precision@10}$ (left), $\overline{Global - Recall}$ (center) and $\overline{Jaccard - Similarity}$ (right) for the initial (hollow markers) and last generations (filled markers) of queries evaluated on the testing set (averaged across five runs on 25 topics).

*description of the topic?* (**RQ1**), *Are the queries evolved by the proposed strategies effective when tested for the same topic but on a new* corpus? *In other words, are the proposed strategies able to avoid overfitting?* (**RQ2**), and *What are the trade-offs among different MOGP strategies for topical search when attempting to address conflicting objectives such as high precision and recall?* (**RQ3**).

The comparison presented in Fig. 7 offers an answer to research question **RQ1** by contrasting the results obtained by the first generation of queries and by the queries evolved by each of the evaluated strategies. Note that the first generation of queries is the same for all the assessed strategies, resulting in the same performance.

Firstly, the results suggest that, with the exception of **Co6**, all the other analyzed strategies demonstrate statistically significant improvements in $\overline{Precision@10}$ when evaluating the queries from the final generation on the test set, in comparison to those queries generated directly from the topic description. A statistically significant improvement in $\overline{Global - Recall}$ is evident in **Co3** when compared to **Co1**, **Co5**, **Co6** and **Co7**. Additionally, **Co3** achieves higher $\overline{Global - Recall}$ when compared to **Co2** and **Co4**, although the difference is not statistically significant. A statistically significant loss in diversity (*i.e.*, an increase in $\overline{Jaccard - Similarity}$) is observed in the evolved queries for all the strategies when compared to the queries directly generated from the topic description. **Co3** achieves statistically significant improvements in diversity when compared to **Co1**, **Co5**, **Co6** and **Co7**. Improvements in diversity (although not statistically significant) can also be observed when comparing **Co3** with **Co2** and **Co4**.

Regarding research question **RQ2**, the results show evidence that the answer is positive, since for the majority of the cases the comparison among the strategies when evaluated on the test set provides results consistent with those observed on the training set. All the strategies, with the exception of **Co6** (which appears to experience overfitting), exhibit similar performance in terms of $\overline{Precision@10}$ on the test set. However, as anticipated, the performance is slightly lower than that achieved during the training stage. All the

**Table 2 Best- and worst-performing strategies for each evaluation metric on the training and testing sets.**

| Metric | Training | | Testing | |
|---|---|---|---|---|
| | Best | Worst | Best | Worst |
| $\overline{Precision@10}$ | Co1 Co5 Co6 Co7 | Co3 | Co1 Co2 Co3 Co4 Co5 Co7 | Co6 |
| $\overline{Global - Recall}$ | Co3 | Co6 | Co3 | Co1 Co5 Co6 Co7 |
| $\overline{Jaccard - Similarity}$ | Co6 | Co1 Co5 | Co3 | Co1 Co5 Co6 |

combinations, reach values above 0.8 for $\overline{Precision@10}$ during training, while most of the combinations attained values over 0.5 for the same metric on the testing set.

Similar to what we observe on the training set, the best $\overline{Global - Recall}$ on the test set is obtained by **Co3** followed by **Co2** and **Co4**. Also, in the same way as we observe on the training set, **Co2**, **Co3** and **Co4** achieve very good performance in terms of $\overline{Jaccard - Similarity}$. An interesting result is that **Co1**, **Co5**, **Co6** and **Co7** result in the worst performance in terms of $\overline{Global - Recall}$ and the same behavior is observed for $\overline{Jaccard - Similarity}$. Our analysis reveals that the primary reason for **Co6**'s underperformance on the test set is its susceptibility to overfitting. An examination of the results produced by this strategy indicates that it tends to generate queries that are excessively specific, often returning only a single relevant document. **Co7** significantly mitigates this issue by enhancing **Co6** with the objective of maximizing $|Ret^\star|$.

Table 2 summarizes these comparisons, indicating the strategies with best and worst performance when evaluated on the training and testing sets. We include more than one strategy when these strategies result in a very similar performance. This trade-off analysis provides an answer to research question **RQ3**.

## Comparison of the MOGP strategies against a MOEA strategy

To investigate research question **RQ4** (*How effective are the queries evolved by the proposed strategies in comparison to queries that are built using MOEA strategies?*), we compare the performance of MOGP-Co3 (which showed the best balance of the performance metrics when compared to other MOGP strategies) against the most effective MOEA strategy reported in *Cecchini et al. (2018)*, to which we refer to as MOEIR.

MOEIR is a MOEA for Information Retrieval designed for queries with a straightforward structure consisting of a disjunctive list of terms. The MOEIR strategy attempts to maximize two fitness functions aimed at penalizing queries that do not contribute to diversity. The inspiration for these fitness functions was drawn from biological principles, where a population of queries can be thought of as individuals competing for a finite resource, which, in this context, represents relevant documents. Consequently, the outcomes of these fitness functions for a specific query, denoted as $q_i$, depend not only on the query itself but also on other queries within the same population. To formally define these fitness functions it is necessary to assume a total order on the queries that constitute $P$. The first of these fitness functions, retrospective precision at rank

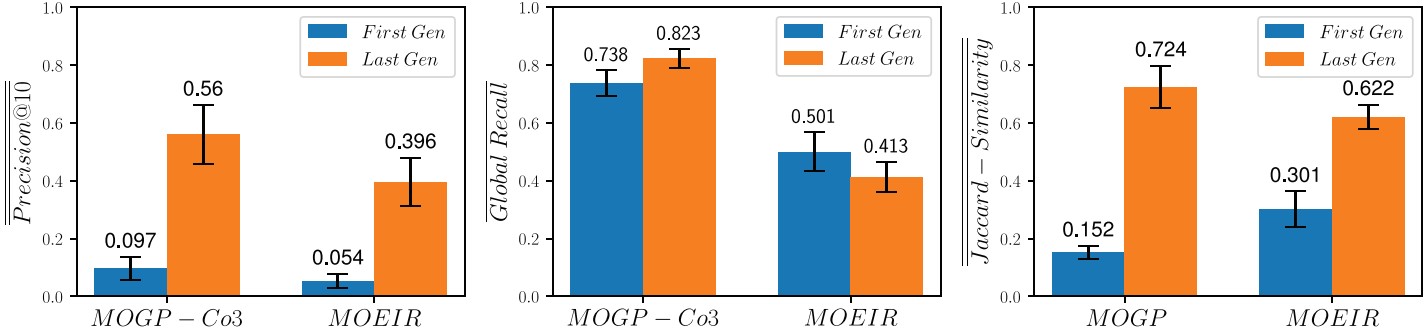

**Figure 8 Comparison of the proposed MOGP-Co3 strategy against the MOEIR (*Cecchini et al., 2018*) on the test set during the first and last generations.**

10, assesses the precision of a query's top ten results while considering the resources (documents) consumed (retrieved) by preceding queries in a population. The second fitness functions, retrospective recall, quantifies how comprehensively a query retrieves relevant documents, considering the resources consumed by earlier queries in the population. Figure 8 compares the performance in terms of $\overline{Precision@10}$, $\overline{Global - Recall}$ and $\overline{Jaccard - Similarity}$ achieved by MOGP-Co3 and MOEIR on the test set during the first and last generations. We observe that MOGP-Co3 outperforms MOEIR in terms of $\overline{Precision@10}$ and $\overline{Global - Recall}$, although it's performance is slightly inferior for $\overline{Jaccard - Similarity}$. Based on these results, we could conclude that, overall, MOGP-Co3 demonstrates better performance than MOEIR.

## Comparison of the MOGP strategies against non-evolutionary term-selection strategies

To investigate research question **RQ5** (*How effective are the queries evolved by the proposed strategies in comparison to queries that are built using state-of-the-art term-weighting schemes?*), we compare the MOGP-Co3 strategies against eleven traditional and state-of-the-art term-weighting schemes. The definitions of the term-weighting schemes used in our analysis are presented in Table 3 and are based on the following notation, adapted from *Lan et al. (2005)* and *Domeniconi et al. (2015)*:

- A denotes the number of documents that belong to class $c_k$ and contain term $t_i$.
- B denotes the number of documents that belong to class $c_k$ but do not contain the term $t_i$.
- C denotes the number of documents that do not belong to class $c_k$ but contain the term $t_i$.
- D denotes the number of documents that do not belong to class $c_k$ and do not contain the term $t_i$.
- N denotes the total number of documents in the collection (*i.e.*, N = A + B + C + D).

**Table 3 Definition of different term-weighting approaches.** Note that some formulations include the expression $\max(X, 1)$ to prevent the possibility of undefined values, such as divisions by zero or $\log(0)$.

| Scheme | Formulation |
|---|---|
| TGF (*Salton & Buckley, 1988*) | $A + C$ |
| IDF (*Salton & Buckley, 1988*) | $\log(N/(A + C))$ |
| TGF* (*Domeniconi et al., 2015*) | $A$ |
| TGF*-IDFEC (*Domeniconi et al., 2015*) | $A \times (\log((C + D)/\max(C, 1)))$ |
| $\chi^2$ (*Quinlan, 1986*) | $N((AD - BC)^2/((A + C)(B + D)(A + B)(C + D)))$ |
| OR (*van Rijsbergen, Harper & Porter, 1981*) | $\log((\max(A, 1) \times D)/\max(B \times C, 1))$ |
| IG (*Quinlan, 1986*) | $(A/N)\log(\max(A, 1)/(A + C)) - ((A + B)/N)\log((A + B)/N) + (B/N)\log(B/(B + D))$ |
| GR (*Quinlan, 1986*) | $IG/(-((A + B)/N)\log((A + B)/N) - ((C + D)/N)\log((C + D)/N))$ |
| $FDD_{0.5}$ (*Maisonnave et al., 2021*) | $(1.25 \times A/(A + C) \times A/(A + B))/((0.25 \times A/(A + C)) + A/(A + B))$ |
| $FDD_{1.0}$ (*Maisonnave et al., 2021*) | $(2.00 \times A/(A + C) \times A/(A + B))/((1.00 \times A/(A + C)) + A/(A + B))$ |
| $FDD_{10}$ (*Maisonnave et al., 2021*) | $(101 \times A/(A + C) \times A/(A + B))/((100 \times A/(A + C)) + A/(A + B))$ |

The eleven term-weighting schemes used for comparison include some traditional information retrieval models that apply unsupervised approaches to determine term importance (no topic labels are used to compute these values), as well as some term-weighting methods that take a supervised approach to assess the importance of a term in a topic or class. The code used for these experiments was adapted from *Maisonnave et al. (2021)* and is available to the research community (https://github.com/ceciliabaggio/term_weighting_methods_full_testing_dataset).

A simple unsupervised term-weighting scheme that we took as a baseline for comparison is based on counting how many documents in the *corpus* contain the given term. This scheme, which is known as term global frequency (TGF), aims to improve recall but typically results in poor precision. Alternatively, other unsupervised term-weighting schemes are based on the premise that common terms are poor discriminators. This is the case of the widely used global factor inverse document frequency (IDF) (*Salton & Buckley, 1988*), which penalizes terms based on the number of documents containing the term. To compare our approach with a basic global weighting scheme we also considered IDF as a baseline.

Other term-weighting methods that we used for comparison take advantage of class-label information, which gives rise to supervised term-weighting schemes. One of such methods is TGF*, which is a supervised variation of TGF that instead of counting the number of documents with the term in the *corpus*, counts the number of documents in a class that contain the term. A limitation of TFG* is that it favors those terms that occur frequently in a class but does not penalize terms that occur in many class. A supervised variant of IDF, known as inverse document frequency excluding category (IDFEC) (*Domeniconi et al., 2015*) penalizes frequent terms but avoids penalizing those terms occurring in several documents belonging to the relevant class. The combination of TGF* with IDFEC gives rise to a supervised term-weighing scheme known as TGF*-IDFEC, which simultaneously accounts for the descriptive and discriminating power of a term with

respect to a relevant class. A traditional supervised term-weighting score used for comparison is given by the conditional probability of a term occurring given a class, which gives rise to the odds ratio (OR) (*van Rijsbergen, Harper & Porter, 1981*). We also took as baseline methods some supervised term-weighting scores derived from information theory (*Shannon, 1948*), including chi-squared ($\chi^2$), information gain (IG) and gain ratio (GR) (*Quinlan, 1986*). Finally, we considered the $FDD_\beta$ score (*Maisonnave et al., 2021*), which is defined as follow:

$$FDD_\beta = (1 + \beta^2) \frac{A/(A + C) \times A/(A + B)}{(\beta^2 \times A/(A + C)) + A/(A + B)}.$$

The $FDD_\beta$ score is a supervised term-weighting scheme that uses a tunable parameter $\beta$ to favor different objectives in the information retrieval task. By using a $\beta$ value higher than 1 we can weight descriptive relevance higher than discriminative relevance while a $\beta$ smaller than 1 weights discriminative relevance higher than descriptive relevance. To carry out our comparative analysis we used $FDD_{0.5}$, $FDD_{1.0}$ and $FDD_{10}$.

To be able to compare the MOGP-Co3 strategy (which showed the best balance of the performance metrics when compared to other MOGP and MOEA strategies) with the schemes mentioned above, we run each method with each of the 25 topics from the dataset. Unlike the MOGP strategies, the query-building process (in the supervised approaches) does not require extra information besides the class-label or topic. This is in contrast with the approach adopted by MOGP, which requires a text snippet (topic description provided by the ODP dataset) to start formulating queries.

Each query built using a term-weighting method consists of the best term (unigram, bigram or trigram) weighted by the scheme at hand. Thus we obtained the first 100 terms ordered by a score, from greatest to smallest. The score represents the weight of the term according to the strategy. Here, the 100 queries are thought to constitute a "population", similar to the ones in the evolutionary approaches. Queries were built based on the vocabulary extracted from the same set of documents used to train the MOGP methods ("training set"), which comprises 2/3 of the dataset. The resulting queries were used to retrieve results from the same dataset that was used to evaluate the MOGP methods ("testing set"), which contains 1/3 of the dataset. The set of matching documents was sorted using query-document TF-IDF similarity and used to compute the performance metrics following the same approach taken by the MOGP methods. For instance, the performance metric $\overline{\overline{Precision@10}}$ looks into the first 10 documents retrieved by each query.

The performance metrics measured on the training and testing sets comparing the term-weighting methods presented in Table 3 against MOGP-Co3 are reported in Figs. 9 and 10, respectively. Note that in these figures we refer to the average (avg) of the metrics. This means that the average in the case of MOGP-Co3 corresponds to the double over-lined averages mentioned before, while in the rest of the methods it represents the single average (single over-lined). The explanation behind this is that in the term-weighting schemes, there is no random seed to start the query generation process, thus we run each

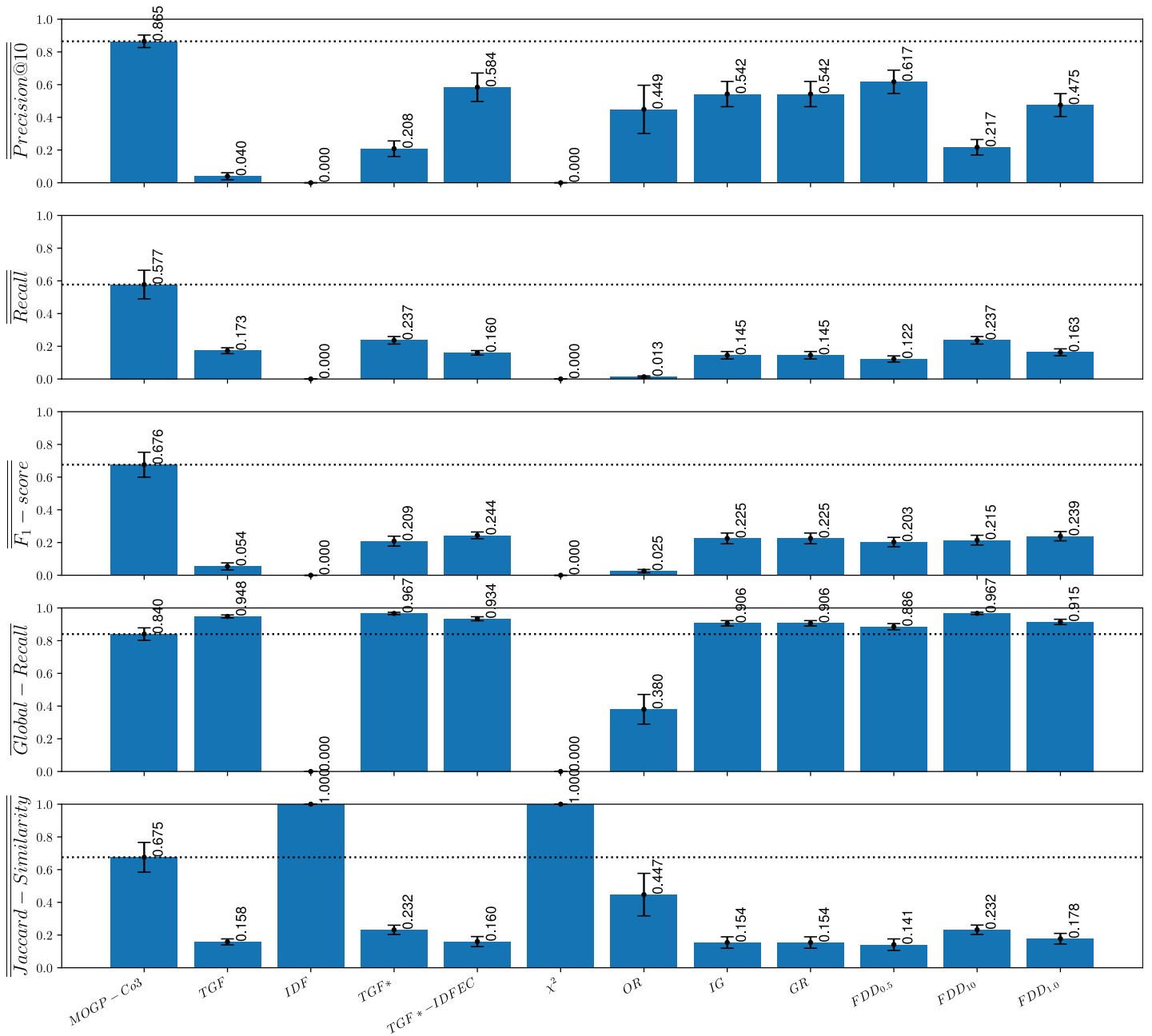

**Figure 9 Performance comparison of the analyzed methods on the training set (performance metrics averaged over 25 topics from the ODP dataset).**

combination of method and topic only once, whereas, in the evolutionary schemes, we completed five runs for each combination.

In general, we observe that MOGP-Co3 has a better performance than the other methods in terms of $\overline{Precision@10}$ and $\overline{Recall}$ (thus in $\overline{F1-score}$) both in the training and testing sets. Also, we observe that the $\overline{Global - Recall}$ score achieved by MOGP-Co3 is competitive, and the result achieved was only slightly worse when compared to the

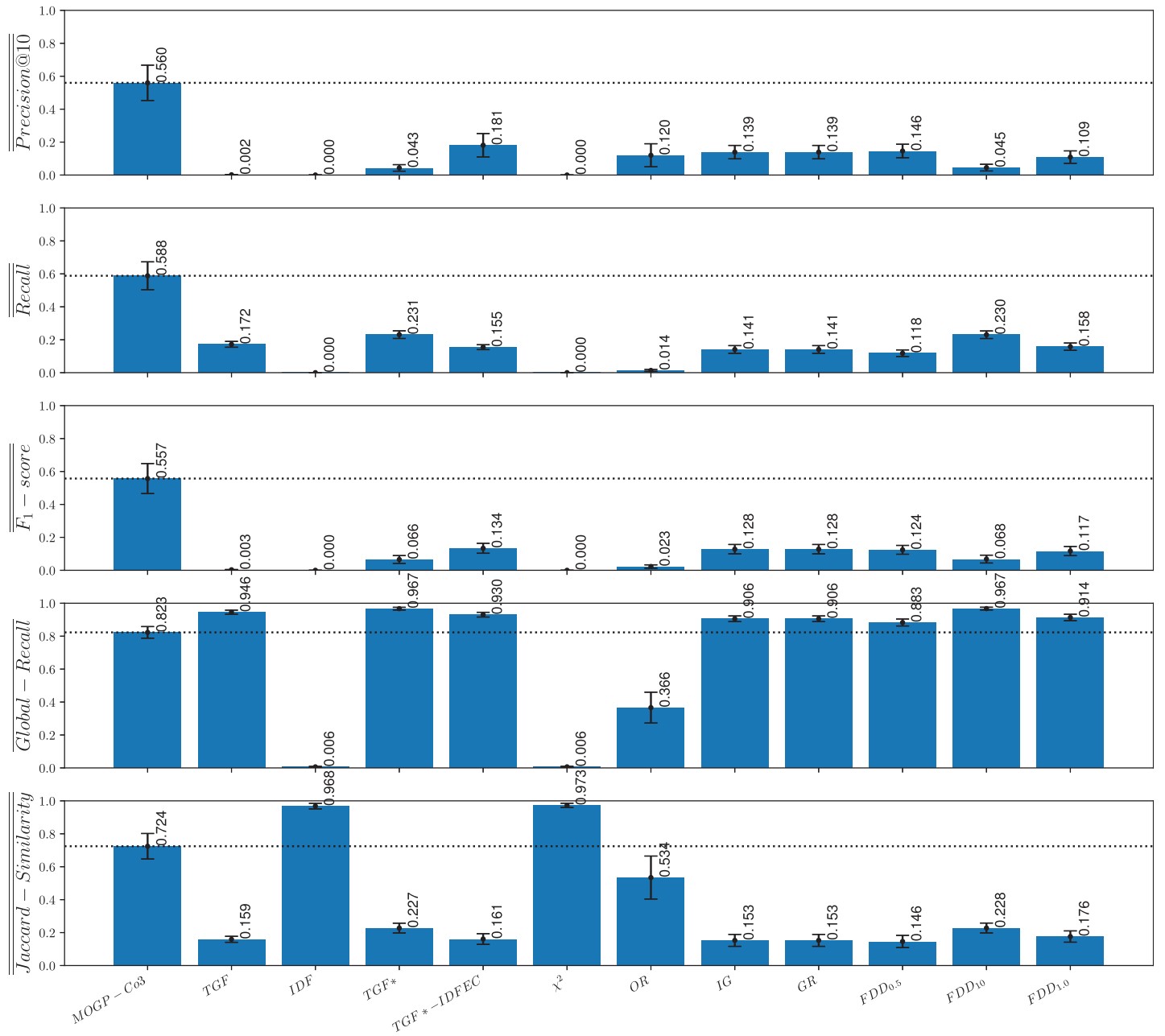

**Figure 10** Performance comparison of the analyzed methods on the testing set (performance metrics averaged over 25 topics from the ODP dataset).

methods with the highest $\overline{Global-Recall}$. Hence, due to the superiority of MOGP-Co3 in terms of $\overline{Precision@10}$ and $\overline{Recall}$, we can claim that MOGP-Co3 has an overall advantage when simultaneously accounting for $\overline{Precision@10}$, $\overline{Recall}$ and $\overline{Global-Recall}$.

On the other hand, we observe that the MOGP-Co3 strategy achieves a higher $\overline{Jaccard-Similarity}$ value than that attained by the term-weighting methods. However, the low $\overline{Jaccard-Similarity}$ obtained by these methods is due to the fact that each of the

built queries is necessarily different from the other queries in the same population, as they are constructed using different terms. In contrast to this, queries evolved by the MOGP-Co3 strategy will tend to share common terms, hence resulting in similar result sets.

## Computational time of the MOGP strategies

The proposed MOGP strategies involve several processes that demand significant computational time. The initialization process involves constructing a population $P$ of Boolean queries, with each query represented as a tree-like individual. The computational time required for building such initial population of queries is $O(|P| \times 2^d)$, with $d$ as the maximum tree depth (in our experiments, we set $d = 17$). Throughout each generation, the computational time is mainly affected by the cost of submitting $|P|$ queries to a search engine, retrieving a document collection for each query, computing the fitness functions associated with the respective MOGP strategy, executing the processes inherent in the NSGA-II algorithm, and managing an external archive of solutions discovered during the evolutionary process. The time required for submitting each query depends on the complexity of the query itself, the search algorithm used by the search interface, and the hardware and infrastructure, among other factors.

The computational time required to compute the fitness functions associated with each objective is highly dependent on each specific function. Global fitness functions are more expensive than local ones. In particular, for the **Co3** strategy, the computation of $Entropic - Precision@10$ and $Entropic - Recall$ have a computational time of $O(|P|)$ and $O(|P| \times |Ret_q|)$, respectively, assuming constant-time relevant membership testing in the relevant set. On the other hand, the **Co1** strategy requires computing $Precision@10$ and $Recall$, with a significantly lower computational time of $O(1)$ and $O(|Ret_q|)$, respectively. Finally, the time complexity of the classical NSGA-II algorithm at each generation is $O(m \times |P|^2)$, where $m$ is number of objectives (*Deb et al., 2002*). A more detailed and recent mathematical analysis of the time complexity of NSGA-II, taking into account enhancements obtained through crossover, is presented in *Doerr & Qu (2023)*.

In our analysis, a typical run of 150 generations with a population of size $P = 100$ requires building and evaluating 15,000 queries, with an average computational time of 2.5 s per query, including the overhead resulting from the various processes inherent to the experiments. For certain MOGP strategies and topics, the cumulative time required for evolving a query population exceeded 10 h based on a standard desktop PC with 16 GB of RAM, SSD, and Intel i7 processor. We anticipate that introducing various optimizations, such as effective caching strategies, can significantly reduce this computational time. However, this is outside the scope of the present work.

It is important to highlight that the process of evolving a population of queries through a MOGP strategy is typically a one-time task performed offline. The optimized queries generated through this process can subsequently be employed for the efficient real-time gathering of information for various purposes, including news and social media monitoring, without the necessity for retraining unless explicitly needed. Consequently, considering the superior performance of certain MOGP strategies in comparison to other approaches and their relevance in applications requiring offline training, we believe that

the answer to research question **RQ6** (*Is the computational time required justifiable based on the effectiveness achieved?*) is affirmative, even considering the time needed to evolve a population of topical queries.

## CONCLUSIONS

This article proposed novel MOGP strategies for topic-based search. The strategies are aimed at learning syntactically rich topical queries with a particular focus on attaining high precision and global recall while preserving diversity. The primary contribution of this study is the development of two fitness functions based on information theory, which can improve recall at the population level and diversity at the phenotype representation level. These fitness functions, which we refer to as $Entropic - Precision@10$ and $Entropic - Recall$, are reformulations of the classical $Precision@10$ and $Recall$ metrics, respectively. Also, we analyzed the effect of directly minimizing the $J - Sim - Index$ between the retrieved results of pairs of queries as an alternative mechanism to seek the maximization of the phenotypic representation diversity.

Seven strategies that result from combining different objectives that seek to maximize performance and preserve diversity were thoroughly examined. The strategies were evaluated using a framework consisting of more than 350,000 webpages classified into 448 topics, specifically designed to evaluate topical search. In addition to assessing the $Precision@10$ score of individual queries, we also evaluated $Global - Recall$ and $Jaccard - Similarity$ at the population level. The analysis was completed both on the training and testing set, allowing to conclude that most strategies do not suffer from overfitting.

Maximizing (any form of) precision and (any form of) recall are commonly seen as conflicting objectives. Our evaluations indicate that maximizing diversity has a higher negative impact on precision than maximizing recall does. This follows from observing that the minimization of $Jaccard - Similarity$ has a consistently negative impact on $Precision@10$ both on the training and testing sets.

Our analysis allows concluding that there is not a single best strategy and that the choice of objectives to be maximized or minimized should be guided by the application at hand. As a general rule, the minimization of $Jaccard - Similarity$ should be avoided if high precision is required. However, if achieving good global recall is important, then the best strategies are those that seek to simultaneously maximize some form of recall and diversity.

The proposed methods have shown an overall superior performance compared to previous related approaches based on MOEA strategies. Our analysis also highlights the advantages and limitations of the MOGP strategies when compared to query construction schemes based on traditional and state-of-the-art term-weighting schemes. In general, we observe that the MOGP strategies achieve higher precision and recall than the analyzed non-MOGP strategies, but are more prone to diversity loss. Another limitation of the MOGP strategies is that evolving a population of topical queries is a computationally expensive process. This work assumes topic-based suggestions are computed offline and has not addressed time and space complexity issues. We recommend the article proposed

by *Lissovoi & Oliveto (2019)* as a reference in the analysis of the time and space complexity of the problem of evolving Boolean conjunctions of $n$ variables by means of GP.

As part of our future work, we plan to investigate other strategies for achieving high global recall and diversity. In particular, we will look into alternative mechanisms for learning new vocabularies not only through supervised methods but also employing semi-supervised and non-supervised approaches. Also, the possibility of incorporating collaborative filtering solutions based on evolutionary algorithms, like those discussed in *Ar & Bostanci (2016)*, will be explored with the goal of designing hybrid topic-based search systems. The *corpus* used in our work is based on ODP, which is currently being continued as the Curlie Project (https://curlie.org/). However, the proposed strategies can be applied to learn topical queries from any collection where documents are classified into topics. In the future, we plan to use other corpora to carry out additional evaluations. Finally, we plan to combine some of the proposed strategies with a multi-population approach such as the one presented in *Cecchini et al. (2018)* to further improve population heterogeneity.

### Funding
This work was supported by CONICET, Universidad Nacional del Sur (PGI-UNS 24/N051 and PGI-UNS 24/N052), and ANPCyT (PICT 2019-03944). The funders had no role in study design, data collection and analysis, decision to publish, or preparation of the manuscript.

### Grant Disclosures
The following grant information was disclosed by the authors:
CONICET, Universidad Nacional del Sur: PGI-UNS 24/N051 and PGI-UNS 24/N052.
ANPCyTL: PICT 2019-03944.

### Competing Interests
Ana G. Maguitman is an Academic Editor of PeerJ Computer Science.

### Author Contributions
- Cecilia Baggio conceived and designed the experiments, performed the experiments, analyzed the data, performed the computation work, prepared figures and/or tables, authored or reviewed drafts of the article, and approved the final draft.
- Carlos M. Lorenzetti conceived and designed the experiments, performed the experiments, analyzed the data, performed the computation work, prepared figures and/or tables, authored or reviewed drafts of the article, and approved the final draft.
- Rocío L. Cecchini conceived and designed the experiments, performed the experiments, analyzed the data, performed the computation work, prepared figures and/or tables, authored or reviewed drafts of the article, and approved the final draft.
- Ana G. Maguitman conceived and designed the experiments, analyzed the data, prepared figures and/or tables, authored or reviewed drafts of the article, and approved the final draft.

## Data Availability

The code used for the evaluation of the MOGP algorithms is available at GitHub and Zenodo:

- https://github.com/ceciliabaggio/mogp_with_terms.

- Baggio, C. (2023). MOGP with terms index (1.0) [Data set]. Zenodo. https://doi.org/10.5281/zenodo.10027031.

- ceciliabaggio. (2023). ceciliabaggio/mogp_with_terms: First Release (mogp-with-terms-v1.0). Zenodo. https://doi.org/10.5281/zenodo.10045189.

The code used for the comparison against queries based on term-weighting strategies is available at GitHub and Zenodo:

- https://github.com/ceciliabaggio/term_weighting_methods_full_testing_dataset.

- ceciliabaggio. (2023). ceciliabaggio/term_weighting_methods_full_testing_dataset: First release (term-weighting-methods-v1.0). Zenodo. https://doi.org/10.5281/zenodo.10066567.

The dataset used for evaluation is a subset of the dataset available at Mendeley:

Lorenzetti, Carlos; Maguitman, Ana; Baggio, Cecilia (2019), "DMOZ 2006 Dataset and its Wikification", Mendeley Data, V1, https://dx.doi.org/10.17632/9mpgz8z257.1.

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
