# Peer review of "Multi-objective genetic programming strategies for topic-based search with a focus on diversity and global recall"

_PeerJ Computer Science, doi:10.7717/peerj-cs.1710_

## Round 0.1 · original submission · Major Revisions

The reviewers find the work interesting, but raise a number of issues that can be considered to improve the paper.

Reviewer 1 ·

Basic reporting

no comment

Experimental design

Concerning the method description:
_authors refer to the Jaccard similarity index but never define it with an equation or text. It could be valuable of readers given that it is used as a fitness function further in the paper;
_the mutation operator description could be more explained: authors set the mutation probability to 0.3, but don't specify if the probability is applied to a solution or a leaf;
_a summary of NSGA-II selection process could be added;

Validity of the findings

*Comparison among MOPG strategies:
_it is necessary to introduce each of the combination, and explain why authors make these choices;
_|A*| is used in the combination Co7 but only defined line 576;
_line 534 : "On the other hand, the MOEA strategy based on NSGA-II achieves an improvement of 3516% on P@10 and an improvement of 1073% on R over the baseline method" : it seems to be results of the method, but it's not clearly highlighted. I recommand to synthesize results on a Table, allowing to compare all the methods mentioned in paragraph with the new approach. In particular, it is necessary to compare the new method to Cecchini et al. 2018 on the same datasets to show the improvements clearly.
_authors do not specify the hardware and the average execution time of the algorithm.

More generally, this section have to be to be redesigned.

Additional comments

The article is well written (excluding experimentation parts), and the level of language is appreciable. The topic is also very interesting.

Reviewer 2 ·

Basic reporting

The works of Holland (1975, Adaptation in Natural and Artificial Systems) are often cited as the inspiration for evolutionary algorithms, and John Koza for their application in searching for programs represented by trees (Genetic Programming, 1992).

Instead of referring to the values of objectives, particularly when those values are interdependent, the term "phenotype" (e.g., line 303) is typically used to refer to the expression of decision variables in the environment (in this case, the phenotype would be the set of retrieved documents). In the context of the article, a distinction between the two would be beneficial.

Experimental design

When presenting the research questions addressed in the article (lines 111–122), they all revolve around the question of determining which criteria are most effective for query retrieval. If we limit the questions to whether certain additional criteria can be added, it is too similar to what was done in Cecchini 2018, which already provides fitness functions based on the population. Considering that one of the main innovations of the article compared to previous work is the ability to construct queries with a more complex structure (Boolean expressions), one of the research questions should be about the performance improvement gained by this capability (which the results tend to show, by the way). Therefore, I suggest that you add a question in this regard.

Except for mentioning that the initial queries are constructed from terms extracted from the topic description (line 406) and the existence of a size limit, there are no details provided on how they are constructed. However, the performance of evolutionary methods in general and genetic programming in particular is significantly influenced by the design of the initial population. A good sampling of the search space is not always easy to achieve in genetic programming (see, for instance, Schweim, Wittenberg, and Rothlauf's article on sampling error in genetic programming in Nat. Comput. 21, 173–186 (2022)). I suggest that you explain the procedure followed to create the initial queries as a result.

It is unclear why, given a sufficiently expressive representation, a single query could not include all the data on a particular topic (line 158). You should provide justification for this claim.

You mention that you tested different mutation rates. It would be interesting to demonstrate how this rate affects performance.
There should be a description of how the ranking of the retrieved documents is determined when presenting the fitness functions you use (after line 458).
(Line 472) The population size should be written as |P| rather than |P| + 1.
When I first read the equation, I did not understand why n was raised to the power of 10, and the notation n_i10 (line 478) can be a bit confusing.
The comparison is based on 25 topics x 5 runs (line 503). First of all, why are there only 25 topics? Additionally, you only present the total variability. Without knowing the variability of the results for each topic, it is difficult to assess if the number of runs is sufficient. You should justify this choice or conduct more experiments.
The selection of objective combinations seems arbitrary. Some combinations do not appear (e.g., EP@10 + R), and others make us question why they are included (e.g., Co7). This choice should be justified.
The section (line 526) comparing the strategies of the article with those of other articles by the authors is difficult to follow. It is not clear what is being compared to and based on what criteria. A table or a graph would be more suitable to present the results during the discussion.
You mention that the random solutions have low precision (line 556). However, the graph in Figure 4 suggests otherwise.
The accent on precision should be removed from the labels of the graphs (Figure 4). I wonder if it is truly relevant to plot the Pareto boundaries; they do not contribute much to visualization and, on the contrary, tend to show that Co1 is better than Co3, which is not true when considering the population as a whole.
On what criteria are you basing the statement that Co3 is preferable (line 567), considering it performs worse on 2 out of 3 comparison criteria, unless the GR criterion is considered the most important for retrieval?
The strategy Co3 is compared (line 659) with criteria that are not used in the comparison of MOGP strategies among themselves. What is the reason for using recall (average) if the population should be considered as a whole?

Validity of the findings

The model training is highly supervised; it requires the topic description and a set of relevance-indexed documents for that topic. The method seems to be heavily dependent on the dataset. It is unfortunate that it has only been tested on a single dataset. If possible, to confirm the usefulness of the method, experiments should be conducted on other datasets.
The computation time (or time complexity) is a critical factor for studying evolutionary methods. It is important to provide at least an estimation of the computation time required to obtain the results.

Additional comments

The article is interesting, well-written, and well-structured, with a comprehensive literature review providing context and results that appear to confirm the usefulness of the proposed methods. However, there are some issues that need to be addressed:
1) It is necessary to present more detailed results, including the number of runs, the number of topics, and the exploration of other datasets.
2) Efforts should be made to provide clearer explanations and interpretations of the results to enhance their understanding.
3) It is essential to provide information on computation time or time complexity to determine whether the method is applicable in real-world use cases.

Addressing these issues will strengthen the overall quality and applicability of the research presented in the article.

Reviewer 3 ·

Basic reporting

The paper is well-written and easy to read. The following suggestions may help improve the manuscript.

1. Related Works section can be switched with the Background section, since the latter discusses some of the methodologies.
2. What is |A^\star| in line 525 of CO7?
3. Legends and tick labels of the figures are too small.
4. Please add references for all the fitness functions. This is not clear which ones are lifted from other papers and which ones are not.
5. The fitness function using entropy is one of the main contribution. Did the authors come up with the formula? If not, please cite the reference.
6. Add a column for the references for each of the scheme in Table 3.

Experimental design

The paper highlights the use of GP and entropy fitness functions as the main contributions. I have some questions regarding the methodology.

1. How are the parameters in Table 1 selected?
2. The results does not show how GP (using structured Boolean queries) outperforms GA (using unstructured query). I recommend that the authors perform more simulations to emphasize the difference.
3. I suggest including GA in the methods for comparison.
4. It will be interesting to add the worst MOGP strategy (Co6) in the comparison. If (Co6) can outperform other methods, then emphasize this in the text. If not, mention that the choice of fitness function is important.
5. There are already several improved version of NSGA-II in the literature. I suggest that the authors consider other more recent multi-objective optimization algorithms which have built-in strategies that balances diversity and good fit.
6. It is mentioned in line 286 that "we evaluate parameter settings based on classical GP settings". If classical GP settings are used, why is this a new contribution?

Validity of the findings

The authors performed some tests based on different metrics. The following suggestions and questions might help improve the manuscript.

1. Use also the best and worst metrics (not just the average) in comparison.
2. It is unclear whether the code for the proposed method is publicly accessible or if only the comparison codes are available. I recommend that the authors include a subsection explaining the functionality of the code, enabling other users to reproduce the results.
3. The metrics (precision, recall, similarity) used for comparison are also the fitness functions, which the algorithm tries to optimize. Is the comparison fair if the other methods used different fitness functions?

Additional comments

This is an interesting and relevant research. I am looking forward to reviewing the revised manuscript.

---

## Round 0.2 · accepted · Accept

The reviewers recommend that the paper be accepted for publication. Please make the minor changes suggested by reviewer #2.

Reviewer 1 ·

Basic reporting

The authors have taken due account of the various comments made by the reviewers. The rewriting work is substantial, and brings many clarifications to the article. As requested, the experimentation section has been completely reworked, and provides a better understanding of the method's benefits.

Experimental design

No comment

Validity of the findings

No comment

Additional comments

No comment

Reviewer 2 ·

Basic reporting

no comment

Experimental design

no comment

Validity of the findings

no comment

Additional comments

The revised article addresses positively all the issues I raised in the initial submission.

Here are some minor corrections that should improve the article:

In Line 450, it is noted that the maximum depth of queries in the initial population is 17, but this is incoherent with Table 1, which states the initial tree depth is between 1 and 5.

Line 453 is greater than the text width.

The results on the complete training set (25 topics) are never shown. The same figure as Figure 7 for the training set should be interesting.

Reviewer 3 ·

Basic reporting

No comment.

Experimental design

No comment.

Validity of the findings

No comment.

Additional comments

The authors have addressed all my comments and recommendations.